# The Effect of Countermeasure Readability on Security Intentions

Tim Smit [1], Max van Haastrecht [1,2,*] and Marco Spruit [1,2,3]

1 Department of Information and Computing Sciences, Utrecht University, Princetonplein 5, 3584 CC Utrecht, The Netherlands; t.k.smit1@uu.nl (T.S.); m.r.spruit@lumc.nl (M.S.)
2 Leiden Institute of Advanced Computer Science (LIACS), Leiden University, Niels Bohrweg 1, 2333 CA Leiden, The Netherlands
3 Department of Public Health and Primary Care, Leiden University Medical Center (LUMC), Albinusdreef 2, 2333 ZA Leiden, The Netherlands
* Correspondence: m.a.n.van.haastrecht@liacs.leidenuniv.nl

**Abstract:** Human failure is a primary contributor to successful cyber attacks. For any cybersecurity initiative, it is therefore vital to motivate individuals to implement secure behavior. Research using protection motivation theory (PMT) has given insights into what motivates people to safeguard themselves in cyberspace. Recent PMT results have highlighted the central role of the coping appraisal in the cybersecurity context. In cybersecurity, we cope with threats using countermeasures. Research has shown that countermeasure awareness is a significant antecedent to all coping appraisal elements. Yet, although awareness plays a key role within the PMT framework, it is generally challenging to influence. A factor that is easy to influence is countermeasure readability. Earlier work has shown the impact of readability on understanding and that readability metrics make measuring and improving readability simple. Therefore, our research aims to clarify the relationship between countermeasure readability and security intentions. We propose an extended theoretical framework and investigate its implications using a survey. In line with related studies, results indicate that people are more likely to have favorable security intentions if they are aware of countermeasures and are confident in their ability to implement them. Crucially, the data show that countermeasure readability influences security intentions. Our results imply that cybersecurity professionals can utilize readability metrics to assess and improve the readability of countermeasure texts, providing an actionable avenue towards influencing security intentions.

**Keywords:** protection motivation theory; cybersecurity countermeasures; readability; structural equation modeling; theory of planned behavior

## 1. Introduction

Online presence is greater than ever. According to a conservative estimate, as of July 2021, there were 4.8 billion active internet users worldwide. The global internet population has grown by more than 257 million over the past year [1]. Especially since the outbreak of COVID-19, companies and public institutions were pushed to adapt their (business) strategy to a more digital one [2–5]. With that, people's likelihood of becoming a victim of cybercrime increased.

McAfee reported worldwide losses from cybercrime in 2020 amounted to just under $1 trillion, an increase greater than 50 percent from 2018 [6]. Besides monetary loss, there are other costs like opportunity costs, system downtime, brand damage and loss of trust, and employee morale damage [6]. Therefore, it is pivotal that SMEs are aware of their risks related to data protection, privacy, and cybersecurity, and receive help in minimizing them. Available solutions are abundant, but these do not meet the needs of for instance small- and medium-sized enterprises (SMEs) that have no proficiency in information technology (IT) or resources to invest in complex and expensive applications [7,8].

Application developers often look to motivate their users to steer them towards favorable security behavior (SB). This is a response to the common proposition that humans are the weakest link in the security chain [9]. To do this, theories and findings in the field of behavioral cybersecurity should be applied. Looking at the issue from both a protection motivation theory (PMT) and the theory of planned behavior (TPB) perspective, self-efficacy is often found to be a strong predictor of security behavior [10,11].

Concisely, self-efficacy in the security context can be seen as "the feeling of being able to implement the protection methods" [12]. Hanus and Wu [10] concluded that, within PMT, self-efficacy is the strongest antecedent of security behavior. Therefore, it is imperative to (textually) formulate proposed countermeasures in such a way that self-efficacy in users of cybersecurity assessment applications is stimulated optimally.

Increasing readability could very well be a key to persuading users to implement these cybersecurity controls, as readability positively influences people's understanding of information security policies [13,14]. Several widely used readability formulas can be applied to give feedback on the readability of countermeasures before being exposed to end-users. However, this approach currently lacks a scientific ground, as behavioral security research does not take into account the relationship between countermeasure readability and security intentions [15–17].

PMT provides a tried and tested framework that can be used to explain behavior in the security context [10]. This paper, therefore, aims to analyze and clarify the relationship between countermeasure readability and behavioral intention through the lens of PMT (extended by TPB) in the cybersecurity context. This work seeks to provide an answer to the following question: how does the readability of cybersecurity countermeasures influence security intentions? In answering this research question, we contribute an extension to the PMT framework. Additionally, by quantifying the impact of countermeasure readability, we open up an actionable path towards positively influencing security intentions.

Answering the main research question involves building up a theoretical framework from relevant findings in the literature (Section 2), which then serves as a foundation for the conceptual framework presented in Section 3. Based on these sections, Section 4 discusses the developed research methodology. Section 5 describes our analysis and its results. The results are followed by an elaboration of the findings in the discussion in Section 6, giving a display of interpretations, implications, limitations, and recommendations for future work. We conclude our paper in Section 7, providing a summary of our research.

## 2. Theoretical Background

Self-efficacy (SE) is defined as "beliefs in one's capabilities to mobilize the motivation, cognitive resources, and courses of action needed to meet given situational demands" [18]. Self-efficacy has been argued as the most focal or pervasive mechanism of human agency which motivates and regulates individual behavior [19].

When people have a high level of SE, they have a stronger internal conviction about their ability to mobilize the motivation, cognitive resources, and courses of action needed to successfully execute the task at hand [20]. It influences the amount of effort, self-regulation, and the initiation and persistence of coping efforts when facing obstacles (relating to the task at hand) [21].

Determinants of SE are performance accomplishments, vicarious experiences, verbal persuasion, and physiological feedback. Performance accomplishments are the most important factor in SE and refer to experiences of executing similar behaviors or tasks. If a person performed well previously in a similar task, they are more likely to feel confident about the current task.

Vicarious experience refers to derived confidence from other people (that one can relate to) carrying out a task. If one sees a peer failing at a task, this negatively affects his/her SE. Verbal persuasion describes encouragement or discouragement to one's performance or capability to perform a task. The effect of this factor is dependent on the credibility of the one persuading.

Lastly, physiological feedback refers to sensations from a person's body and how this is interpreted by that person. For instance, one could sense a tension in the stomach after realizing that their operating system is prone to security breaches due to a lack of updates. Interpreted as anxiety, this will negatively impact SE. Conversely, when this is interpreted as excitement, the person's SE will be increased in the task at hand [22]. The task being updating the system with the latest security patches.

The concept of SE is an important construct in several theories on human behavior and motivation; examples being protection motivation theory (PMT) [23], self-efficacy theory of motivation [21], self-determination theory [24], and health belief model (HBM) [25]. Studies applying HBM and PMT to the security context, have pointed out that SE is a strong determinant of computer SB [26,27] and desktop SB [10], respectively. Other studies showed a significant relationship between SE and intention to information system security policy compliance [11] and intention for compliant SB [28].

The three domains, computer SB, desktop SB, and information system security policy compliance, can be paralleled to the more general idea of users' integration of cybersecurity controls into their SB. Bauer and Bernroider [28] showed that in this general sense, SE had a significant effect on user intentions for compliant SB. The next subsection will elaborate on these theories applied to this context and what role SE plays in them.

## 2.1. Protection Motivation Theory, Theory of Planned Behavior, and Self-Efficacy

Protection motivation theory (PMT) comes from research on fear appeals, which is mainly centered around how fear-arousing communication affects behavior [23]. In brief, PMT assumes that human behavior is formed through a cost–benefit analysis where risks associated with certain behavior are held up against the costs of trying to eliminate or mitigate these risks [29].

PMT distinguishes multiple components within fear appeals. This makes it possible to determine common variables that impact attitude, which affects behavioral change [30]. These components are one's cognitive processes and are defined by coping appraisal (perceptions of the recommended coping response to the threat in question) and threat appraisal (perceptions of how threatened one believes they are). The two components each have their antecedent (coping awareness and threat awareness, respectively) and are both split up into multiple factors.

Hanus and Wu [10], looking at the impact of users' security awareness on desktop security behavior from a PMT perspective, adjusted the model to fit their context. That is, they translated coping awareness to countermeasure awareness (CA). Results demonstrated that there was support for their suggested model (Figure 1).

A few years prior, Ifinedo [11] saw an overlap between the theory of planned behavior (TPB) and PMT and sought to integrate these to help explain users' compliance in information system security policy. Later, Martens et al. [12] also synthesized PMT and TPB, showing elements of TPB (mainly subjective norm) are valuable to add to PMT.

TPB is based on the premise that, when behaving in certain ways, individuals make logical, reasoned decisions by evaluating the information available to them [31]. TPB states that intention is the primary antecedent of planned behaviors. Intention is influenced by three determinants, attitude toward a behavior, subjective norm, and perceived behavioral control.

Fishbein and Ajzen [32] define subjective norm as follows: "[someone's] perception that most people who are important to him or her think he or she should or should not perform the behavior in question". Perceived behavioral control represents "the person's belief as to how easy or difficult performance of the behavior is likely to be" and is based upon SE as introduced by Bandura in 1977 [22,33].

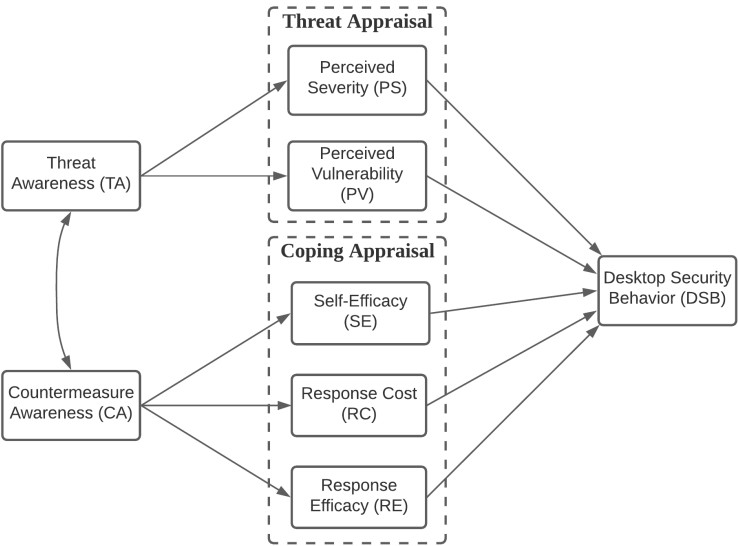

**Figure 1.** Protection motivation theory applied to desktop security behavior. Redrawn from Hanus and Wu [10].

## 2.2. Text Readability

Richards and Schmidt [34] explain that reading is the process of perceiving a text to understand its content. Readability is what causes some texts to be easier to read than others.

There is a clear distinction with legibility, which is concerned with the visual perception of layout and typeface [35]. Dale and Chall [36] define readability as follows. "The sum total (including all the interactions) of all those elements within a given piece of printed material that affect the success a group of readers has with it. The success is the extent to which they understand it, read it at an optimal speed, and find it interesting".

Klare [37] offers a more concise definition: "the ease of understanding or comprehension due to the style of writing". This definition, along with others [38,39], emphasize vocabulary and sentences, noting that these are the primary sources of reading difficulty.

Research has shown that an increase in readability causes better comprehension, retention, reading speed, and perseverance [35]. However, there are antecedents to readability too. These can be divided into two dimensions, the reader and the text itself.

Text features that impact readability are content, style, design, and organization and belong to the internal dimension. On the reader's part, prior knowledge, reading skill, interest, and motivation affect the ease of reading. Prior knowledge, interest, and motivation are related to the subject of the text, whereas reading skill captures a reader's overall reading proficiency in the language at hand [35]. Interest and motivation can be related to a reader's disposition or inclination to reading a certain text.

### 2.2.1. Measuring Text Readability

People have sought to measure the textual dimension of readability. Predicting the difficulty of texts has many useful applications and is therefore widely used in publishing, education, healthcare, and the military. Even courts accept their use in testimonies [35]. Therefore, this has been an area of research since the late nineteenth century.

A century later, over 200 readability formulas had been constructed for different languages and validated each with their study [35]. These metrics take in two or three textual features (e.g., number of words, syllables, complex words, or characters), which are negatively correlated with a text's readability. A major advantage of these formulas is that they can be applied to a text immediately, without the need for a corpus, training of models, and expert knowledge, as would be the case if a natural language processing approach was taken [40].

2.2.2. Limitations of Readability Metrics

There are a few limitations to traditional readability metrics. Firstly, in general, they focus solely on internal features of the texts in question and ignore the familiarity (or lack thereof) that readers might have with terms (prior knowledge) [41].

Secondly, they do not take semantics into account. This means texts could have issues concerning their meaning, affecting their readability. Thirdly, these formulas presuppose people are similar in interest, motivation, maturity, reading skill, and characteristics [35,42]. Fourthly, formulas do not score textual organization and design [35].

Lastly, they are language-specific, being optimized only for one language [43]. This implies usage for other languages is not possible without tweaking and tuning them.

Fortunately, the field of natural language processing (NLP) is catching up, meaning algorithms with higher accuracy have already emerged. These can take into account as many as 70 features and can classify sentences far more precisely.

However, it should be noted that work by Dell'Orletta et al. [40] concluded that basic features, such as sentence length and word length, are still the most important in both sentence and document readability classification. Nonetheless, since traditional formulas take in basic internal features (e.g., average sentence/word length or average syllables per word), measurements become increasingly inaccurate as text length decreases.

2.2.3. Readability of Cybersecurity Countermeasures

Research into the effect of readability in the cybersecurity domain is scarce. Yet, earlier work exists and has formed an inspiration for our approach. Alkhurayyif and Weir [13] compared the eight most popular traditional readability metrics against manual human comprehension metrics in information security policies (ISPs) and demonstrated a correlation between human and computer metrics.

They pose that since "readability has an impact upon understanding ISPs, [...] the application of suitably selected readability metrics may allow policy designers to evaluate their draft policies for ease of comprehension prior to policy release". They mention that "there may be grounds for a readability compliance test that future ISPs must satisfy". In a later paper, Alkhurayyif and Weir [14] further examine the relationship between readability and ISP understanding, concluding that readability has a clear influence. As ISPs include cybersecurity controls and cover similar material, applying readability metrics to countermeasures might be beneficial.

## 3. Research Model and Hypotheses

A couple of conclusions can be derived from Section 2. Firstly, PMT and TPB apply to the context of cybersecurity safeguards. Moreover, they can be synthesized.

This research takes the approach of Martens et al. [12], adding 'subjective norm'. As with others [10,44–49], in this research, 'attitude toward protective behavior' is left out, meaning the predictive value of the framework on security intentions (SI) is examined directly.

Within this model, SE was shown to be a strong predictor multiple times [10,11]. SE is likely correlated with countermeasure readability (CR) since it seems evident that the extent to which one comprehends a task influences their confidence in doing what is needed to aptly cope with a cyber threat.

For response cost (RC), likewise, it seems clear that task comprehension affects one's perception of the mental effort it takes to perform that task. Since implementing countermeasures is the coping behavior for PMT in the cybersecurity context, CR can only be conceptually linked to constructs that fall under coping appraisal.

Nevertheless, a relationship between CR and RE will not be hypothesized, as it is not evident from both literature and common sense that CR affects one's perception of how well implementing a security control will guard them against a cyber threat. SE and RC are therefore chosen to be the points of attachment CR will have in the extended PMT framework (Figure 2). Text readability has two dimensions: the human side and the textual side. For the human side, DuBay [35] mentions four antecedents for a text's readability.

Two of these, interest and motivation, are captured by the latent construct countermeasure reading disposition (CRD). Another antecedent is prior knowledge. However, because of the strong conceptual overlap with CA, prior knowledge is not seen as a separate construct in the model, but seen as congruent with CA.

Reading skill will not be taken up in the framework since an assessment thereof is outside of the scope of this research and would make the questionnaire too long. Five readability formulas will be used to capture the internal dimension of readability.

Section 2.2.2 mentioned some drawbacks of traditional readability metrics. For instance, their language dependence. From this, it follows that this work will be done in the context of the English language only. However, for the proposed conceptual model, these drawbacks mainly imply that CR is a latent construct measured by an operationalization of perceived readability.

These conclusions lead to the conceptual framework and hypotheses of Figure 2.

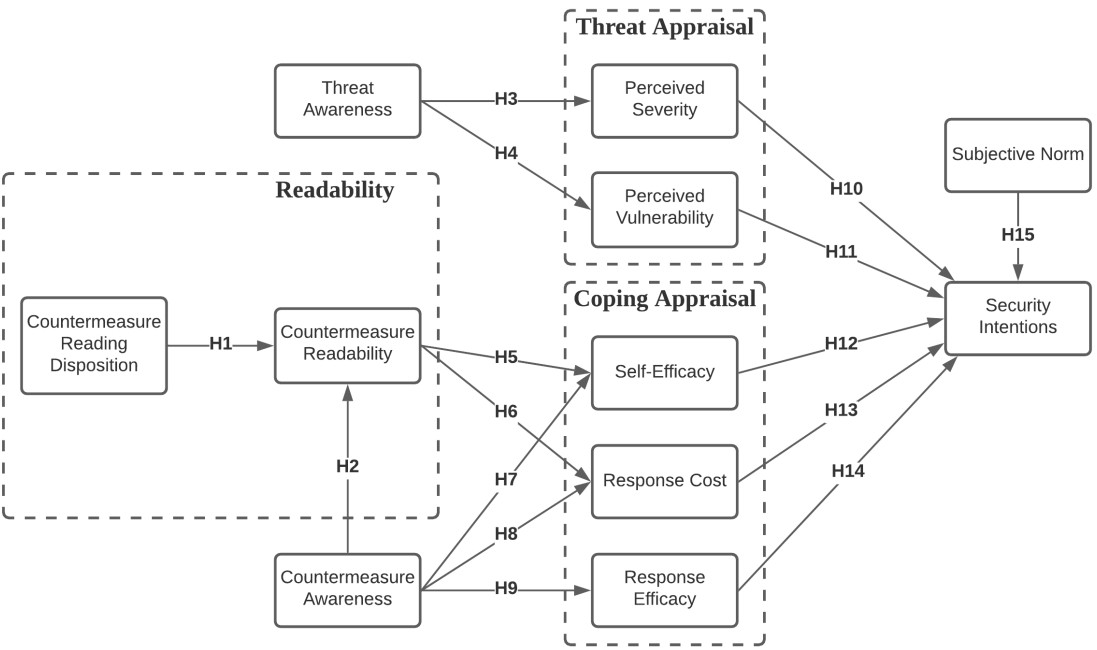

**Figure 2.** Proposed conceptual framework. An extended PMT model applied to a cybersecurity context based on Hanus and Wu [10].

Most of our conceptual framework has already been studied in earlier work. In this paper, the readability section is sought to be added to the PMT/TPB model to explain the effect of CR on SI. Hypotheses proceeding from the PMT/TPB variables will be based on the literature and therefore only be aimed to be confirmed.

Ultimately, this research aims to find a relation between CR and security intentions through SE and RC. Therefore, SI and CR are the main dependent and independent variables in the research model, respectively.

Intention is taken rather than actual behavior, as behavior is difficult to measure or observe in this research context. Fortunately, a strong and consistent relationship between the two constructs has been established before [50,51].

Our model contains multiple exogenous and endogenous variables. Table 1 provides an overview of all relationships and corresponding hypotheses along with their sources. Hypotheses H1 and H2 are deduced from work done by DuBay [35]. These hypotheses will be tested since DuBay [35] did not quantitatively demonstrate these relations.

Relationships denoted by H5 and H6 are the main focus of this study and are sought to be established. The other hypotheses are previously confirmed by the respective studies mentioned in Table 1.

**Table 1.** List of hypotheses this paper investigates.

| Index | Hypothesis | Source(s) |
|---|---|---|
| H1 | Countermeasure reading disposition positively affects countermeasure readability | [35] |
| H2 | Countermeasure awareness positively affects countermeasure readability | [35] |
| H3 | Threat awareness is a positive predictor of perceived severity | [10,12] |
| H4 | Threat awareness is a positive predictor of perceived vulnerability | [48] |
| H5 | Countermeasure readability is a positive predictor of self-efficacy | Proposed |
| H6 | Countermeasure readability is a negative predictor of response cost | Proposed |
| H7 | Countermeasure awareness is a positive predictor of self-efficacy | [10,12] |
| H8 | Countermeasure awareness is a negative predictor of response cost | [10] |
| H9 | Countermeasure awareness is a positive predictor of response efficacy | [10,12] |
| H10 | Perceived severity is a positive predictor of security intentions | [28,44,45,52] |
| H11 | Perceived vulnerability is a positive predictor of security intentions | [44] |
| H12 | Self-efficacy is a positive predictor of security intentions | [10,28,45,52] |
| H13 | Response cost is a negative predictor of security intentions | [44,45,48,49,52] |
| H14 | Response efficacy is a positive predictor of security intentions | [10,28,44,52] |
| H15 | Subjective norm is a positive predictor of security intentions | [12,49] |

## 4. Methodology

### 4.1. Questionnaire

Based on the literature discussed in previous sections, a structured questionnaire was drawn up. It started with a brief overview, including the goal of the study, what was expected of the participants in the online questionnaire, and an informed consent form.

After respondents gave consent, they were introduced with a case description. This is also known as a vignette and its usage is recommended by [53]. Participants were asked to project themselves into this hypothetical situation to minimize faulty data that could result from respondents feeling disassociated from the context. Applied to our context, this could mean that cybersecurity threats were not being felt or seen. This is included, as the application of PMT requires participants to recognize being exposed to a threat [23].

A feeling of indifference about the statements, for example, could arise otherwise, which would yield distorted data. It also provides for more alignment between the participants, concerning how they approach the questions and statements. Lastly, it straightens out possible differing associations participants may have with the statements.

In the questionnaire, participants are shown four cyber threat definitions and countermeasures against them. The selection of threats (Appendix A) and corresponding controls are based on the most common cyber threats as recognized by Cisco Systems [54].

Respondents were asked to respond to statements about both threats and controls related to all constructs found in the conceptual framework (Section 3). Statements corresponding to the constructs within PMT are based on work by Hanus and Wu [10], Martens et al. [12], and Anderson and Agarwal [55]. All statements were to be rated using a 7-point Likert scale, ranging from 'strongly disagree' to 'strongly agree' to make the questions uniform across the measured constructs. This was in line with Vagias's [56] proposed usage of Likert scales.

what is important to highlight are the constructs SE, SI, and CR, as these are the focus of this study. SE was operationalized through two statements per countermeasure. The first is "I feel confident carrying out these countermeasures". The statement is a derivative of Hanus and Wu's [10] operationalization and is based on SE's definition worked out in Section 2.1. "Carrying out these countermeasures is easy" is the second questionnaire item used for measuring SE. This one is based on the application of PMT for cybercrime in the work of Anderson and Agarwal [55], which is also used by Martens et al. [12]. Work by Liu et al. [57] could also be used to support this operationalization, as they conclude that perceived difficulty is a dimension of SE. SI was inquired through one statement per countermeasure, e.g.: "I will carry out the countermeasures against malware".

The four cybersecurity safeguards each had two versions, one 'hard' version according to the five readability metrics (found in Appendix A), and likewise one 'easy' version. This was done to create more varied data concerning the variable CR. They were also used as a 'sanity check' on the variable CR, to see whether 'hard' texts were actually perceived as harder (see Section 5). Online tools enabled convenient and fast calculations of the readability scores [58,59].

A respondent was randomly appointed either a 'hard' or an 'easy' version. These four safeguards were then shown in random order between participants (evenly distributed), so decreased attention would not affect one safeguard that would be the last one shown each time. The questionnaire closed with two text entries, one for joining the gift card draft, the other for any comments the respondents may have had.

### 4.2. Participants

Participants were found using convenience and snowball sampling. Some participants were personally asked to participate, others through WhatsApp groups. A second platform used to draw participants was the Microsoft Teams channel of the bachelor's course Information Security, taught at Utrecht University. As a reward, we included a EUR 15.00 gift card draft (i.e., a EUR 15.00 gift card was randomly allotted to one participant).

Taking PMT into the questionnaire meant the number of questions became quite high. Harrison [60] advises against a questionnaire that is too long because respondents are less likely to answer (compared to a short one) and often pay less attention. For that reason, further questions that were not essential (e.g., demographic) to this research, but would increase the questionnaire's length, were excluded.

This gained 131 respondents after running the survey for 14 days. This number was reduced to 88 ($N = 88$) after checking the responses' validity using three measures.

First, the 'red herring' question had to be answered correctly (see Appendix A). This question asked participants to fill the gap (multiple choice) in the previously given definition of a man-in-the-middle attack, which was essential to understand since statements about it had to be responded to later.

Second, all questions needed to be answered. Third, the survey had to be completed in a credible time span (>5 min). With an average duration of 15.80 min and a standard deviation of 10.25 min (*Median* = 13.04) when removing outliers, this duration threshold was seen as reasonable. Outliers were defined by data points with greater duration than one hour, as these subjects had most likely paused between questions (some even resumed after days, causing enormous outliers). If a record failed to meet one of these criteria, it was not taken into the results.

The total sample was composed of mostly non-native English speakers (97%). Three people reported English as their first language, 82 noted it was their second language, two called it their third language, and one respondent described English as a fourth language.

Of the 88 subjects, 77 were Dutch speakers (first language). The other 11 subjects reported having the following languages as their first tongue: 1 Arabic, 4 Indonesian, 1 Russian, 1 Swahili, and 1 Tamil. Of these 85 non-natives, 59 reported their English reading level was roughly on par with or above their reading level in their native tongue.

This implies that at least 26 out of 88 participants have an English reading level below their first language reading level. Expectedly these reported, on average, a score significantly lower ($M = 5.50$, $SD = 0.91$) on reading skill than the other 62 ($M = 6.21$, $SD = 1.03$), $t(52.91) = -3.22$, $p = 0.002$.

Reading skill was measured by a 7-point Likert scale (ranging from 'strongly disagree' to 'strongly agree') response to the statement "I can understand academic English texts". Of the subjects, 61 (69%) were between 18 and 24 years old (most likely being students due to convenience and snowball sampling), 11 were between ages 25 and 34, seven were between 55 and 64, five were 65 or older, two between 45 and 54, and the remaining two respondents fell into the age groups under 18 and 35–44 years old, respectively.

### 4.3. Formative Measurement Model Evaluation

As this work sought to supplement PMT, the data resulting from the questionnaire had to be analyzed using a suitable method. The partial least squares structural equation modeling (PLS-SEM) method is the perfect fit for this research, as it allows for the estimation of causal systems that include latent variables that can be endogenous, exogenous, or both [61]. Moreover, Hanus and Wu [10], Ifinedo [11], and Bauer and Bernroider [28] use this method. All calculations concerning the PLS algorithms were done in R using the SEMinR package [62]. The link to the code and data used for the analysis can be found in Appendix D.

This research's conceptual model translated into a formative measurement model in PLS-SEM. Each construct was considered a latent variable since they could not be directly measured but merely estimated by an operationalization of the concept. This operationalization resulted in manifest variables (a.k.a. indicators in SEM) that measured, or 'formed,' the latent constructs.

Contrary to reflective models, in formative models evaluating the internal consistency of constructs is inappropriate. Diamantopoulos [63] stresses, when "formative measurement is involved, reliability becomes an irrelevant criterion for assessing measurement quality". Criteria, such as Cronbach's $\alpha$ and composite reliability, were therefore not applicable to this research.

In a critical assessment of the usage of PLS-SEM Hair et al. [64] find inter alia that researchers do not employ and sometimes even misuse criteria for model evaluation. Considering this, this study will be attentive to their proposed guidelines, elaborated upon by Hanafiah [65].

Within PLS-SEM, there are certain recommended methods for model assessment. For formative outer models, there are three criteria to assess. First, multicollinearity, measured by the variance inflation factor (VIF). Second, the indicators' relative contribution to their constructs, noted by their weights. Third, construct validity, estimated by the significance of weights and convergent validity.

Since formative measurement model construction should be primarily founded on theory rather than statistics [66,67], the latter two of these criteria do not have a fixed value that must be achieved. Hair et al. [64] propose important statistics concerning these criteria should at least be reported.

#### 4.3.1. Variance Inflation Factor

Before the significance of outer weights of the measurement model could be analyzed and relationships could be interpreted, the formative measurement model had to be assessed for collinearity issues. The first step in this process was detecting collinearity; that is, looking for high correlations between two formative indicators [61].

High correlations were not expected, since PMT is well-tested and its constructs and their operationalizations (which output indicators/manifest variables) are distinct and well delineated. Moreover, operationalizations were based on other work, where they were shown to be valid [10,12,55]. Manifest variables are the basis of the measurement model and, therefore, SEM analysis as a whole. As a result, high collinearity between them would be problematic.

To assess collinearity between formative indicators, the variance inflation factor was calculated, as recommended by Ramayah et al. [68]. Hair et al. [61] state that if VIF $\geq$ 5, it is likely a collinearity issue is at play. The VIF values for the manifest variables did not surpass 4.671 and averaged at 2.062, safely failing to exceed the recommended threshold.

#### 4.3.2. Size and Significance of Indicator Weights

The original sample ($N = 88$) was resampled using the commonly used bootstrap technique [69]. This was done to test the significance of path coefficients as well as weight significance. Following the recommendation of Hair et al. [61], the number of bootstrap subsamples was set to 5000.

Regarding indicator weights, ultimately, this process resulted in few significant weights. Only ta3, cr1, re1, ps2, se3, and sn1 were found to be significant. The affixes 1, 2, and 3 stand for malware, phishing, and MitM, respectively. The indicator weights, when significant, imply that a measured variable explains a significant share of the variance in the latent formative construct [70].

Despite the fact that many indicator weights were not significantly related to the latent construct, they were not dropped. The rationale behind this was the fact that these variables each contribute conceptually to their formative constructs. This was further substantiated by the indicators being based on both theory regarding the concepts [35] and their operationalizations [10,12,55].

Moreover, previous research [47,71] also retained the indicators with insignificant weights for the same reason. This argument was also supported by Edwards and Bagozzi [66] and Petter et al. [67], as they pose that though statistical considerations should be taken into account, conceptual reasoning has more weight in determining whether or not to leave out formative indicators. Appendix B displays a complete overview of all indicator weights.

### 4.3.3. Convergent Validity

An assessment of convergent validity looked at whether observed variables measuring the same construct were related [72]. For this research, the constructs CRD, CA, CR, and SE were measured using different aspects of those concepts. The other constructs measured the same aspect for the four different cyber threats.

Table 2 presents the correlations between these constructs' indicators and Appendix A provides exact statements that were used for each of the indicators. For a complete overview of all correlation values, please consult Appendix C.

**Table 2.** Correlations of formative indicators. Note that correlation is a bidirectional relationship.

| Construct | Relationship | Correlation |
|---|---|---|
| Countermeasure reading disposition | crd1 → crd2 | 0.739 |
| Countermeasure readability | cr1 → cr2 | 0.773 |
| Countermeasure readability | cr3 → cr4 | 0.733 |
| Countermeasure readability | cr5 → cr6 | 0.818 |
| Countermeasure readability | cr7 → cr8 | 0.869 |
| Self-efficacy | se1 → se2 | 0.654 |
| Self-efficacy | se3 → se4 | 0.532 |
| Self-efficacy | se5 → se6 | 0.592 |
| Self-efficacy | se7 → se8 | 0.649 |

This section assessed the measurement model, which gave insight into the validity of the measured constructs and their indicators. The formative measurement model that was put forth was found to be adequately valid, mainly due to theoretical grounds.

## 5. Results

### 5.1. Comparing Perceived Readability to Readability Metric Scores

For perceived readability, two operationalizations were used, "This text is easy to read" and "This text is easy to understand". An independent-samples t-test was conducted to compare the readability between 'hard' ($n = 180$) and 'easy' ($n = 172$) countermeasure texts. Table 3 delineates a significant difference in the perceived readability for 'hard' and 'easy' for both operationalizations.

**Table 3.** *t*-Test results comparing readability metrics with perceived readability and corresponding descriptive statistics.

|  | This Text Is Easy to Read | | This Text Is Easy to Understand | |
|---|---|---|---|---|
| *M* | 5.47 ('hard') | 5.95 ('easy') | 5.36 ('hard') | 5.84 ('easy') |
| *SD* | 1.29 ('hard') | 1.06 ('easy') | 1.33 ('hard') | 1.17 ('easy') |
| *df* | 350 | | 350 | |
| *t* | 3.9 | | 3.6 | |
| *p* | <0.001 | | <0.001 | |
| Skewness | −1.28 | | −1.19 | |
| Kurtosis | 1.40 | | 0.84 | |

*5.2. Research Model Results*

In this study, a PLS-SEM analysis tested all posed hypotheses. Its results are visually displayed in Figure 3.

The main focus of this work was to check whether CR influences SI. To assess this accurately, PMT was enhanced with a readability section and applied to the cybersecurity context. The primary endogenous (or dependent) variable was security intentions. Ultimately, the model explained 58% of its variance ($R^2 = 0.577$).

$R^2$ indicates the model's predictive power [73]. $R^2 = 0.577$ falls well within standards for behavioral studies [74,75] and is considered a moderate effect size by Hair et al. [61]. Moreover, it was higher than the $R^2$ Hanus and Wu [10] ($R^2 = 0.461$) and Martens et al. [12] ($R^2 = 0.28$) achieved.

Besides security intentions, other $R^2$ values were deemed weak (RE) or very weak (PS, PV, and RC). This indicates that the variance in those variables was not well explained by the constructs anteceding them. CR, on the other hand, did receive a high $R^2$ value, with $R^2 = 0.552$. Finally, 57% ($R^2 = 0.573$) of the variance in self-efficacy was explained by countermeasure readability and countermeasure awareness.

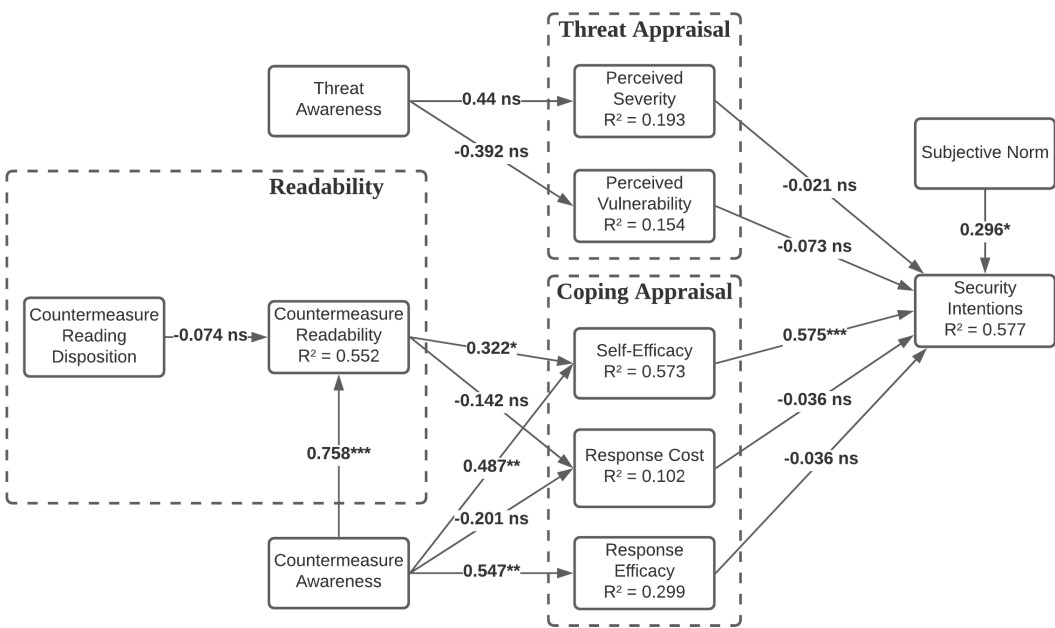

**Figure 3.** The research model results. Note: $^*p < 0.05$; $^{**}p < 0.01$; $^{***}p < 0.001$; *ns*—not significant.

Figure 3 corresponds with the structural, or inner, model of the PLS-SEM model. After having analyzed the outer model in Section 4.3, the structural model was analyzed. The path coefficients the PLS algorithms produced can be interpreted as regular beta coefficients.

Meaning in this case, that if SE changes by one standard deviation, SI changes by 0.567 ($\beta = 0.567$) standard deviations [76].

Significance and confidence intervals were estimated for path coefficients using the bootstrapping method. This resulted in six hypothesized relationships being confirmed through significance.

Countermeasure awareness was found to be a strong significant positive predictor of countermeasure readability ($\beta = 0.758$, $p < 0.001$; H2). Countermeasure readability positively predicted self-efficacy ($\beta = 0.322$, $p < 0.05$; H5). Countermeasure awareness was a strong positive predictor for both self-efficacy ($\beta = 0.487$, $p < 0.01$; H7) and response efficacy ($\beta = 0.547$, $p < 0.01$; H9).

Self-efficacy was found to be a strong significant positive predictor of security intentions ($\beta = 0.575$, $p < 0.0001$; H12). Lastly, for the variable subjective norm a significant positive relationship ($\beta = 0.296$, $p < 0.05$) with security intentions was established, supporting H15. Table 4 displays the findings of the PLS-SEM analysis of the structural model concerning the hypotheses.

**Table 4.** Overview of PLS-SEM results of structural model analysis.

| Hypothesis | Path Coefficient | T Stat. | $F^2$ | 2.5%CI | 97.5%CI | Supported? |
|---|---|---|---|---|---|---|
| H1: CRD→CR | −0.074 | −0.671 | 0.020 | −0.223 | 0.205 | No |
| H2: TA→PS | 0.440 | 1.386 | −0.491 | 0.240 | 0.706 | Yes |
| H3: TA→PV | −0.392 | −1.235 | −0.614 | 0.181 | 0.487 | No |
| H4: CR→SE | 0.322 | 1.892 | 0.047 | 0.138 | 0.719 | No |
| H5: CR→RC | −0.142 | −0.522 | −0.714 | 0.015 | 0.372 | Yes |
| H6: CA→CR | 0.758 | 10.047 | 0.611 | 1.184 | 0.899 | No |
| H7: CA→SE | 0.487 | 2.764 | 0.071 | 0.223 | 0.770 | Yes |
| H8: CA→RC | −0.201 | −0.743 | −0.703 | 0.033 | 0.410 | No |
| H9: CA→RE | 0.547 | 2.928 | 0.080 | 0.426 | 0.760 | Yes |
| H10: PS→SI | −0.021 | −0.139 | −0.257 | −0.005 | 0.343 | No |
| H11: PV→SI | −0.073 | −0.685 | −0.223 | −0.008 | 0.206 | No |
| H12: SE→SI | 0.575 | 3.563 | 0.192 | 0.190 | 0.794 | Yes |
| H13: RC→SI | −0.096 | −0.803 | −0.340 | 0.007 | 0.129 | No |
| H14: RE→SI | −0.036 | −0.277 | −0.210 | −0.014 | 0.297 | No |
| H15: SN→SI | 0.296 | 2.308 | 0.042 | 0.179 | 0.542 | Yes |

## 6. Discussion

The main research question this study aims to answer is: how does the readability of cybersecurity countermeasures influence security intentions?

The results of the analysis of our framework confirm that countermeasure readability affects security intentions through self-efficacy since H5 and H12 are supported. The study demonstrates countermeasure readability is significantly predicted by countermeasure awareness (H2), meaning the more familiar one is with security measures, the easier the measure will be perceived to read.

H1 was not supported, implying that a reader's disposition to reading about countermeasures does not affect their perceived readability. The data suggest countermeasure readability is a critical determinant of self-efficacy (H5). This means that when people are exposed to security controls that help them safeguard against a cyber threat, they feel more confident applying these when they perceive them to be easy to read.

In line with previous research [10,45,52], the results indicate that self-efficacy is a significant predictor of security intentions (H12). When individuals believe or feel confident they can implement a countermeasure, they are likely to have the intention to carry out the security measure.

Countermeasure readability does not predict response cost (H6 was rejected); that is, the readability of a presented countermeasure does not affect how much mental or

financial effort people think implementing the measure takes. As a result, countermeasure readability only influences security intentions through self-efficacy.

Confirmation of H7 and H9 mean awareness of countermeasures is a significant antecedent of the coping appraisal mechanisms self-efficacy and response efficacy. The awareness of threats was not found to be a significant determinant of perceived severity (H3) and perceived vulnerability (H4). More specifically, this implies that evoking aware-ness of cyber threats or appealing to fear is not enough to steer people to desired security behavior if they do not know of countermeasures or do not know how to counteract cyber threats. This conforms to findings by Hanus and Wu [10], Johnston and Warkentin [77], and Rhee et al. [78].

Concerning the last supported hypothesis, the data point out that subjective norm is a significant determinant for security intentions (H15). This signifies that one is more inclined to act safely regarding cyber threats and measures when people (that individual relates to) think they should adopt protective security behavior.

Finally, the proposed model explains 58% of the variance in the variable SI. According to Hair et al. [61], this is a moderate effect size. $F^2$ indicates an effect for 11 of the 15 re-lationships (Table 2). $F^2$ measures the change in $R^2$ (on a dependent variable) when an independent variable is left out of the model. Cohen [79] proposes $F^2$ values of 0.02, 0.15, and 0.35 account for small, medium, and large effect sizes, respectively.

Contrary to other research (e.g., [12,44,48] relationships concerning threat appraisal constructs are not supported (H3, H4, H10, H11). However, this is in line with work by Hanus and Wu [10] and Tsai et al. [49], suggesting PMT is strongly context-dependent as all related work displayed in Figure 1 differ slightly in their contextual application. Section 6.1 delves into other possible reasons for this lack of support.

Besides the model, Section 5.1 demonstrated there is a significant difference in per-ceived readability between 'easy' and 'hard' countermeasure texts, as classified by five readability metrics (see Appendix A). The distributions of the indicators for countermea-sure readability are highly negatively skewed. This explains why the means of 'hard' and 'easy' countermeasures do not differ much.

As a result, it can be concluded that readability metric scores of textual readability in-deed affect the perceived readability of cybersecurity countermeasure texts. Consequently, if a countermeasure text scores low on readability according to a readability formula it will be perceived as poorly readable, and vice versa.

### 6.1. Implications and Limitations

The results of this research provide empirical ground for the usage of readability metrics to improve countermeasure readability in cybersecurity assessment applications.

Previous work has shown a strong significant relationship between intentions and actual behavior [50,51]. Additionally, countermeasure readability indirectly positively predicts security intention. Hence, it is worthwhile to implement readability checks to improve security control texts before presenting them to users. These metrics are easily taken advantage of since well-documented open source packages are available [80,81]. Table 5 provides example improvements based on readability metric scores.

The first point of critique one may have is on the analysis of the readability metrics' effect on perceived readability (CR in this research). As was shown, the distribution of the measured variables is non-normal, by which the use of a *t*-test could be questioned. After all, *t*-tests assume normality. However, a paper by Lumley et al. [82] demonstrates that *t*-tests are valid with sufficiently large sample sizes. They state that "sufficiently large" is under 100. The sample size for comparing the countermeasures texts was 352, 172 for 'easy' measures and 180 for 'hard' ones. The use of an independent-sample t-test in this situation is therefore justified.

On average, both 'easy' and 'hard' countermeasures were rated relatively high on readability. Synthesized with the strong relationship between CA and CR, this indicates that the subjects were familiar with security jargon terms and cybersecurity in general. This

fits with the fact that part of the sample was taken from a bachelor's class on information security. This is a plausible cause for the strong negative skew reported in Section 5.1.

**Table 5.** Example countermeasure readability improvements. The gray rows contain the more readable countermeasures.

| Threat | Countermeasure | Metric Scores |
|--------|----------------|---------------|
| Malware | Whenever possible, verify updates are automatically being installed. Alternatively, ensure all software packages are updated to their most recent version. Activate the operating system's integrated firewall before you connect any device to the internet or other networks. Employ a virus protection program that frequently performs full system scans optimized for malicious software detection. Safeguard critical data by planning a regular off-site backup (e.g., cloud or tape backup) of systems, so recovery from cyber attacks will be a less complicated process. | FRE = 29.9 FKGL = 12.0 FOG = 14.4 DC = 10.6 CLI = 17.8 ARI = 11.3 |
| Malware | Where possible, make sure that updates are automatically installed. Otherwise, make sure to update all your software to the latest version. Make sure all your connections to the internet (or other networks) go through a firewall. Make sure you have set up an antivirus program that scans your systems for malware. Make sure your important data is regularly backed up in the cloud or another safe place, so coming back from an attack will be easier. | FRE = 61.2 FKGL = 8.5 FOG = 11.9 DC = 7.1 CLI = 12.3 ARI = 8.7 |
| MitM | Verify the establishment of an HTTPS connection when submitting confidential information to a website, signified by the lock symbol in or around the URL bar in the internet browser. Do not transmit confidential information via an insecure public Wi-Fi network. Ensure security of your local network by deploying an intrusion detection system. | FRE = 27.7 FKGL = 13.8 FOG = 17.6 DC = 10.8 CLI = 16.0 ARI = 12.8 |
| MitM | Make sure websites where you give personal info have an HTTPS connection as shown by the lock icon in or around the address bar in your browser. Never give out personal data over an insecure public Wi-Fi network. Make sure your network is secured with an intrusion detection system. | FRE = 61.3 FKGL = 8.8 FOG = 11.5 DC = 7.0 CLI = 11.3 ARI = 8.6 |

Furthermore, this research does not conclude anything about the extent to which readability formulas indicate countermeasure readability (nor readability in general) or how they integrate with the proposed model. That is, readability metrics were not taken into the model, thus refusing to estimate any effect a text's readability score may have on any of the constructs in the model.

Regarding the proposed model, 9 out of 15 hypotheses are not supported. This could indicate that the sample is not homogeneous. Furthermore, Section 4.2 points out that the sample likely contained a high percentage of bachelor students. Hanel and Vione [83] note that "generalizing from students to the general public can be problematic when personal and attitudinal variables are used". Though the bootstrap procedure with 5000 iterations takes advantage of the central limit theorem [84], it does not counter poor population representativeness in the original sample.

Another explanation of the low number of supported hypotheses is the operationalization of the concerning variables. That is, the use of a vignette caused some confusion for respondents. Some reported they would have answered differently if they were not asked to imagine themselves as a restaurant owner with several information systems.

Moreover, it is possible that respondents forgot to answer from the context of the given case description. Different interpretations on how to respond to the statements in the survey can lead to opposing relations being measured, leading to great variance without regression between constructs.

This could be the case with H4 and H11. For instance, if someone is unfamiliar with a cyber threat, they might have perceived themselves as being vulnerable to it, since they did not have the chance to respond to it yet (resulting in a negative relation between TA and PV). When the constructs are seen anteceding one another, as PMT was intended, someone who is well aware of a threat can see themselves as being vulnerable to it (resulting in a positive relation between TA and PV).

Likewise for H11, if one perceives himself or herself as being highly vulnerable he or she may employ countermeasures (resulting in a positive relation between PV and SI). On the other hand, people might rate themselves as not vulnerable to a cyber threat, because they have already implemented security controls.

For formative indicators in general, it is important to capture a concept as completely as possible by measuring different dimensions of a construct [61]. Using multiple items to estimate a latent construct allows for stronger construct validity [85]. However, adding reflective indicators or more formative items comes at a cost. Response rates, completion rates, and data quality suffer from increased questionnaire lengths [60,86,87].

This paper aimed to capture protection motivation processes for cyber threats and measures in general, not only malware, for example. Therefore, constructs were surveyed using the same general statements to capture the essence of a construct concerning four cyber threats/measures. This leads to suboptimal construct validity, which explains the low $R^2$ value for some of the endogenous constructs.

Looking ahead, NLP techniques could be a valuable alternative in assessing counter-measure texts. Countermeasure texts can be short, often containing only a few sentences. Formulas that consider only two or three internal features of a text are inaccurate in these situations. Dell'Orletta et al. [40], speaking of NLP readability assessment, conclude that "assessing the readability of sentences is a complex task, requiring a high number of features, mainly syntactic ones".

This research recommends readability metrics as an opportunity to improve readability in security applications. However, it acknowledges the metrics' limitations. Therefore this work promotes investigating NLP techniques to increase readability in countermeasure texts. Extant research already shows NLP's possibilities and applicability in this respect [88–90]. Traditional metrics, on the other hand, are encouraged to be used as a guideline as soon as possible.

## 7. Conclusions

Stimulating secure behavior is critical in security assessment applications, as human failure in this regard is a primary contributor to cybercrime [9,91].

Work by Alkhurayyif and Weir [13] points to readability metrics for countermeasure text assessment to improve users' compliance to controls. By analyzing cybersecurity protection motivation through the lens of an extended PMT model, this paper has clarified how countermeasure readability influences security intentions.

A structural equation modeling approach to the analysis of the model (Section 3) showed that countermeasure readability affects security intentions through self-efficacy. The data analysis, together with extant literature (see Table 1), provide support for the proposed theoretical model.

This study affirms and contributes to the current body of knowledge by confirming the model's usefulness in the cybersecurity context, showing countermeasure readability has its place in forming protection motivation.

The methodological choices concerning the sample and survey were constrained by the pressure to limit questionnaire length, as this leads to lower response rates inter

alia [60,87]. Nevertheless, our analysis demonstrated the statistical significance of several key coefficients.

Our results highlight the importance of considering readability metrics for assessing countermeasure texts. The improvement of countermeasure readability is perhaps the most accessible avenue towards influencing security intentions. The future is bright in this regard. The use of novel NLP techniques will facilitate more accurate estimations of readability, opening up further possibilities for stimulating secure behavior [40,88–90].

**Author Contributions:** Conceptualization, T.S., M.v.H., and M.S.; methodology, T.S. and M.v.H.; software, T.S.; validation, M.v.H. and M.S.; formal analysis, T.S.; investigation, T.S. and M.v.H.; data curation, T.S.; writing—original draft preparation, T.S.; writing—review and editing, M.v.H. and M.S.; visualization, T.S.; supervision, M.v.H. and M.S.; project administration, T.S. and M.v.H. All authors have read and agreed to the published version of the manuscript.

**Funding:** This work was made possible with funding from the European Union's Horizon 2020 research and innovation program, under grant agreement no. 883588 (GEIGER). The opinions expressed and arguments employed herein do not necessarily reflect the official views of the funding body.

**Informed Consent Statement:** Informed consent was obtained from all subjects involved in the study.

**Data Availability Statement:** The data generated by this research are publicly available at: https://github.com/TKForgeron/SEMinR-Analysis-The-Effect-of-Countermeasure-Readability-on-Security-Intentions (accessed on 14 September 2021).

**Conflicts of Interest:** The funders had no role in the design of the study; in the collection, analyses, or interpretation of data; in the writing of the manuscript, or in the decision to publish the results.

## Abbreviations

The following abbreviations are used in this manuscript:

| | |
|---|---|
| ARI | automated readability index |
| CA | countermeasure awareness |
| CLI | Coleman–Liau index |
| CR | countermeasure readability |
| CRD | countermeasure reading disposition |
| DC | New Dale–Chall readability formula |
| FKGL | Flesch Kincaid grade level |
| FOG | gunning fog score |
| FRE | Flesch reading ease |
| HBM | health belief model |
| ISP | information security policy |
| MitM | man-in-the-middle |
| NLP | natural language processing |
| PLS | partial least squares |
| PMT | protection motivation theory |
| PS | perceived severity |
| PV | perceived vulnerability |
| RC | response cost |
| RE | response efficacy |
| SB | security behavior |
| SE | self-efficacy |
| SEM | structural equation modeling |
| SI | security intentions |
| SME | small- and medium-sized enterprises |
| SN | subjective norm |
| TA | threat awareness |
| TPB | theory of planned behavior |
| VIF | variance inflation factor |
| ZDE | zero-day exploit |

## Appendix A. Questionnaire

**Table A1.** Overview of all questionnaire items.

| Construct Item | Survey Item | Item Options/ Readability Score | Source(s) |
|---|---|---|---|
| age | How old are you? | *Under 18; 18–24; 25–34; 35–44; 45–54; 55–64; 65+* | |
| english_level | "English is my ______ language." | *First; Second; Third; Fourth (or more)* | |
| language | What is your first language? | *Open text entry* | |
| *RS* | | | |
| rs1 | I can understand academic English texts | *7-point Likert scale (1 = strongly disagree; 7 = strongly agree)* | |
| rs2 | "My English reading level is roughly equal to or above my first language reading level." | *True; False* | |
| *TA* | *All variables measuring TA have the same answer options and structure.* | *7-point Likert scale (1 = strongly disagree; 7 = strongly agree)* | |
| Definition: Malware | Malware is a term used to describe malicious software, including spyware, ransomware, viruses, and worms. Malware breaches a network through a vulnerability, typically when a user clicks a dangerous link or email attachment that then installs risky software. | | Cisco Systems [54] |
| Definition: Phishing | Phishing is the practice of sending fraudulent communications that appear to come from a reputable source, usually through email. The goal is to steal sensitive data like credit card and login information or to install malware on the victim's machine. Phishing is an increasingly common cyber threat. | | Cisco Systems [54] |
| Definition: Man-in-the-Middle | Man-in-the-middle (MitM) attacks, also known as eavesdropping attacks, occur when attackers insert themselves into a two-party transaction. Once the attackers interrupt the traffic, they can filter and steal data. | | Cisco Systems [54] |
| Definition: Zero-Day Exploit | A zero-day exploit is an attack that exploits a previously unknown hardware, firmware, or software vulnerability. Unknown meaning not known by software vendors or security software that relies on finding known patterns or signatures hackers use. | | Dempsey et al. [92], Mulloy [93] |
| ta1 | I am familiar with malware | | Hanus and Wu [10] |
| ta2 | I am familiar with phishing | | Hanus and Wu [10] |
| ta3 | I am familiar with a man-in-the-middle attack | | Hanus and Wu [10] |

**Table A1.** *Cont.*

| Construct Item | Survey Item | Item Options/ Readability Score | Source(s) |
|---|---|---|---|
| ta4 | I am familiar with a zero-day exploit | | Hanus and Wu [10] |
| *PS* | *All variables measuring PS have the same answer options and structure.* | *7-point Likert scale (1 = strongly disagree; 7 = strongly agree)* | |
| ps1 | I think malware is a severe problem | | Martens et al. [12] |
| ps2 | I think phishing is a severe problem | | Martens et al. [12] |
| *PS* | *All variables measuring PS have the same answer options and structure.* | *7-point Likert scale (1 = strongly disagree; 7 = strongly agree)* | |
| ps3 | I think a man-in-the-middle attack is a severe problem | | Martens et al. [12] |
| ps4 | I think a zero-day exploit is a severe problem | | Martens et al. [12] |
| *PV* | *All variables measuring PV have the same answer options and structure.* | *7-point Likert scale (1 = strongly disagree; 7 = strongly agree)* | |
| pv1 | It is likely that I become a victim of malware | | Hanus and Wu [10] |
| pv2 | It is likely that I become a victim of phishing | | Hanus and Wu [10] |
| pv3 | It is likely that I become a victim of a man-in-the-middle attack | | Hanus and Wu [10] |
| pv4 | It is likely that I become a victim of a zero-day exploit | | Hanus and Wu [10] |
| red herring | What concept is best described by the following description? ___________, also known as eavesdropping attacks, occur when attackers insert themselves into a two-party transaction. Once the attackers interrupt the traffic, they can filter and steal data. | Phishing; Malware; Zero-Day Exploit; Man-in-the-Middle Attack | |
| *CRD* | *All variables measuring CRD have the same answer options.* | *7-point Likert scale (1 = strongly disagree; 7 = strongly agree)* | |
| crd1 | I am interested in computer security (countermeasures) | | |
| crd2 | I am motivated to read about computer security countermeasures | | |
| *CR* | *All variables measuring CR have the same answer options and structure. The countermeasure definitions' ('easy' or 'hard') scores are found to the right of the definition.* | *7-point Likert scale (1 = strongly disagree; 7 = strongly agree)* | |
| Measure: Malware (easy) | Where possible, make sure that updates are automatically installed. Otherwise, make sure to update all your software to the latest version. Make sure all your connections to the internet (or other networks) go through a firewall. Make sure you have set up an antivirus program that scans your systems for malware. Make sure your important data is regularly backed up in the cloud or another safe place, so coming back from an attack will be easier. | FRE = 61.2; FKGL = 8.5; FOG = 11.9; DC = 7.1; CLI = 12.3; ARI = 8.7 | Swiss Federal Department of Finance [94] |

**Table A1.** *Cont.*

| Construct Item | Survey Item | Item Options/ Readability Score | Source(s) |
|---|---|---|---|
| Measure: Malware (hard) | Whenever possible, verify updates are automatically being installed. Alternatively, ensure all software packages are updated to their most recent version. Activate the operating system's integrated firewall before you connect any device to the internet or other networks. Employ a virus protection program that frequently performs full system scans optimized for malicious software detection. Safeguard critical data by planning a regular off-site backup (e.g., cloud or tape backup) of systems, so recovery from cyber attacks will be a less complicated process. | FRE = 29.9; FKGL = 12; FOG = 14.4; DC = 10.6; CLI = 17.8; ARI = 11.3 | Swiss Federal Department of Finance [94] |
| *CR* | *All variables measuring CR have the same answer options and structure. The countermeasure definitions' ('easy' or 'hard') scores are found to the right of the definition.* | *7-point Likert scale (1 = strongly disagree; 7 = strongly agree)* | |
| Measure: Phishing (easy) | Make sure your systems only allow strong passwords. If it is possible, use two-factor authentication. This gives extra protection against your account being hacked. Never give out personal data on a website you have reached by clicking on a link in an email or text message. No bank will ever send you an email asking that you change your password or confirm your credit card details. Be on guard if you get emails or texts that ask you to take action with consequences when you do not do what they say (loss of money, criminal charges or legal issues, account or card blocking, missed opportunity, misfortune, etc.). | FRE = 66.4; FKGL = 8.5; FOG = 10.2; DC = 7.6; CLI = 11.2; ARI = 9.2 | Swiss Federal Department of Finance [95] |
| Measure: Phishing (hard) | Enforce strong passwords throughout your information systems. Wherever possible, use two-factor authentication. This offers an additional layer of protection to prevent your account from being hacked. Never divulge confidential information (e.g., passwords or credit card details) on websites you have accessed by following a link in an email or text message. Never would a bank or credit card corporation send you an email requesting that you change your password or verify your credit card details. Be skeptical whenever you receive emails that require action from you and threaten you with consequences if you do not follow their insistence (loss of money, criminal charges or legal proceedings, account or card blocking, missed opportunity, misfortune, etc.). | FRE = 47; FKGL = 10.3; FOG = 12.7; DC = 8.8; CLI = 16.1; ARI = 11.3 | Swiss Federal Department of Finance [95] |

**Table A1.** *Cont.*

| Construct Item | Survey Item | Item Options/ Readability Score | Source(s) |
|---|---|---|---|
| Measure: MitM (easy) | Make sure websites where you give personal info have an HTTPS connection, as shown by the lock icon in or around the address bar in your browser. Never give out personal data over an insecure public Wi-Fi network. Make sure your network is secured with an intrusion detection system. | FRE = 61.3; FKGL = 8.8; FOG = 11.5; DC = 7; CLI = 11.3; ARI = 8.6 | Cisco Systems [54,96], Gontharet [97] |
| Measure: MitM (hard) | Verify the establishment of an HTTPS connection when submitting confidential information to a website, signified by the lock symbol in or around the URL bar in the internet browser. Do not transmit confidential information via an insecure public Wi-Fi network. Ensure security of your local network by deploying an intrusion detection system. | FRE = 27.7; FKGL = 13.8; FOG = 17.6; DC = 10.8; CLI = 16; ARI = 12.8 | Cisco Systems [54,96], Gontharet [97] |
| Measure: ZDE (easy) | Stay up-to-date with the latest patches on all your programs. These patch holes in your security, so the risk of newly found weaknesses being exploited is lower. Create a list of approved applications and limit installations to only these. Try to keep down the number of applications you use, as this reduces the risk of vulnerabilities being found across a wide suite of software. Make sure your computers and networks are being scanned for fishy events by an intrusion detection system. | FRE = 61.6; FKGL = 8.8; FOG = 11.5; DC = 7.9; CLI = 12.5; ARI = 9.5 | Mulloy [93], Finjan [98], Sophos [99] |
| *CR* | *All variables measuring CR have the same answer options. The countermeasure definitions' ('easy' or 'hard') scores are found to the right of the definition.* | *7-point Likert scale (1 = strongly disagree; 7 = strongly agree)* | |
| Measure: ZDE (hard) | Keep updated with the latest security patches on all your applications. Patches repair the vulnerabilities in software and operating systems, reducing the risk of newly discovered vulnerabilities being exploited. Create an application whitelist and restrict installations to these. Restraining your number of applications reduces the likelihood of being vulnerable to cyberthreats throughout your software packages. Have an intrusion detection system monitor your computers and networks for suspicious activity. | FRE = 16.4; FKGL = 14.4; FOG = 19; DC = 9.7; CLI = 22.14; ARI = 15.7 | Mulloy [93], Finjan [98], Sophos [99] |
| cr1 (malware) | This text is easy to read | | |
| cr2 (malware) | This text is easy to understand | | |
| cr3 (phishing) | This text is easy to read | | |
| cr4 (phishing) | This text is easy to understand | | |
| cr5 (MitM) | This text is easy to read | | |
| cr6 (MitM) | This text is easy to understand | | |
| cr7 (ZDE) | This text is easy to read | | |

**Table A1.** *Cont.*

| Construct Item | Survey Item | Item Options/ Readability Score | Source(s) |
|---|---|---|---|
| cr8 (ZDE) | This text is easy to understand | | |
| *CA* | *All variables measuring CA have the same answer options and structure.* | *7-point Likert scale (1 = strongly disagree; 7 = strongly agree)* | |
| ca1 (malware) | I am familiar with the content (terms and concepts) of this text | | Adapted from Hanus and Wu [10] |
| ca2 (phishing) | I am familiar with the content (terms and concepts) of this text | | Adapted from Hanus and Wu [10] |
| ca3 (MitM) | I am familiar with the content (terms and concepts) of this text | | Adapted from Hanus and Wu [10] |
| ca4 (ZDE) | I am familiar with the content (terms and concepts) of this text | | Adapted from Hanus and Wu [10] |
| *SE* | *All variables measuring SE have the same answer options.* | *7-point Likert scale (1 = strongly disagree; 7 = strongly agree)* | |
| se1 (malware) | I feel confident carrying out these countermeasures | | Adapted from Hanus and Wu [10] |
| se2 (malware) | Carrying out these countermeasures is easy | | Adapted from Anderson and Agarwal [55] |
| se3 (phishing) | I feel confident carrying out these countermeasures | | Adapted from Hanus and Wu [10] |
| se4 (phishing) | Carrying out these countermeasures is easy | | Adapted from Anderson and Agarwal [55] |
| se5 (MitM) | I feel confident carrying out these countermeasures | | Adapted from Hanus and Wu [10] |
| se6 (MitM) | Carrying out these countermeasures is easy | | Adapted from Anderson and Agarwal [55] |
| se7 (ZDE) | I feel confident carrying out these countermeasures | | Adapted from Hanus and Wu [10] |
| se8 (ZDE) | Carrying out these countermeasures is easy | | Adapted from Anderson and Agarwal [55] |
| *RE* | *All variables measuring RE have the same answer options and structure.* | *7-point Likert scale (1 = strongly disagree; 7 = strongly agree)* | |
| re1 | Carrying out these countermeasures will protect me against malware | | Adapted from Martens et al. [12] |
| re2 | Carrying out these countermeasures will protect me against phishing | | Adapted from Martens et al. [12] |
| re3 | Carrying out these countermeasures will protect me against a man-in-the-middle attack | | Adapted from Martens et al. [12] |
| re4 | Carrying out these countermeasures will protect me against a zero-day exploit | | Adapted from Martens et al. [12] |
| *RC* | *All variables measuring RC have the same answer options and structure.* | *7-point Likert scale (1 = strongly disagree; 7 = strongly agree)* | |
| rc1 (malware) | Carrying out these countermeasures costs me a lot (e.g., financial or mental effort) | | Adapted from Hanus and Wu [10] |

**Table A1.** *Cont.*

| Construct Item | Survey Item | Item Options/ Readability Score | Source(s) |
|---|---|---|---|
| rc2 (phishing) | Carrying out these countermeasures costs me a lot (e.g., financial or mental effort) | | Adapted from Hanus and Wu [10] |
| rc3 (MitM) | Carrying out these countermeasures costs me a lot (e.g., financial or mental effort) | | Adapted from Hanus and Wu [10] |
| rc4 (ZDE) | Carrying out these countermeasures costs me a lot (e.g., financial or mental effort) | | Adapted from Hanus and Wu [10] |
| *SI* | *All variables measuring CR have the same answer options and structure.* | *7-point Likert scale (1 = strongly disagree; 7 = strongly agree)* | |
| si1 | I will carry out the countermeasures against malware | | Adapted from Martens et al. [12] |
| si2 | I will carry out the countermeasures against phishing | | Adapted from Martens et al. [12] |
| si3 | I will carry out the countermeasures against a man-in-the-middle attack | | Adapted from Martens et al. [12] |
| si4 | I will carry out the countermeasures against a zero-day exploit | | Adapted from Martens et al. [12] |
| *SN* | *All variables measuring CR have the same answer options and structure.* | *7-point Likert scale (1 = strongly disagree; 7 = strongly agree)* | |
| sn1 | People I relate to think I should protect myself from malware | | Adapted from Martens et al. [12] |
| sn2 | People I relate to think I should protect myself from phishing | | Adapted from Martens et al. [12] |
| sn3 | People I relate to think I should protect myself from a man-in-the-middle attack | | Adapted from Martens et al. [12] |
| sn4 | People I relate to think I should protect myself from a zero-day exploit | | Adapted from Martens et al. [12] |

**Appendix B. Indicator Weights**

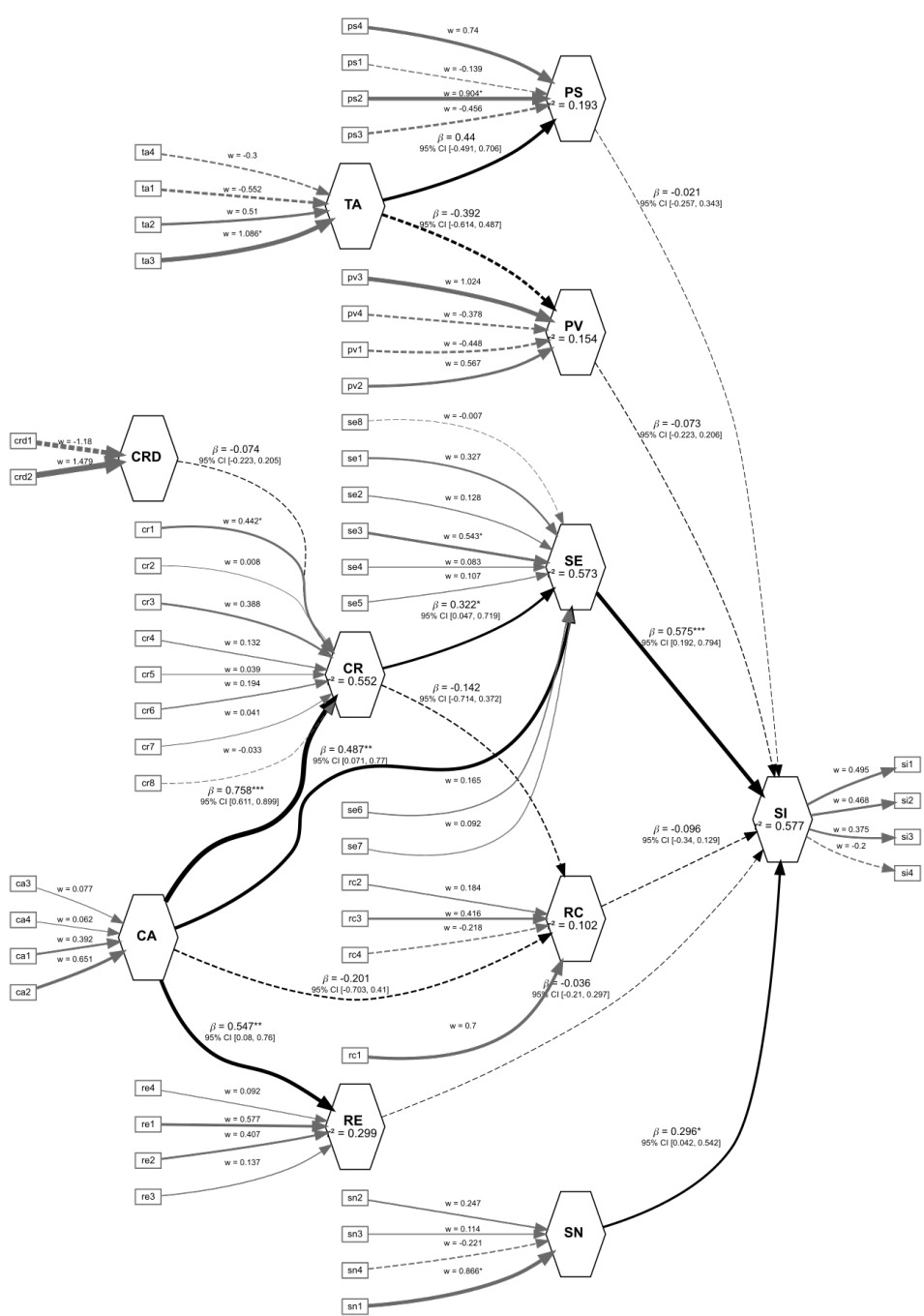

**Figure A1.** A complete overview of the (bootstrapped) PLS-SEM model, including all indicator weights. $^{*}p < 0.05$; $^{**}p < 0.01$; $^{***}p < 0.001$.

# Appendix C. Item Correlations

**Table A2.** Correlations between factors.

|  | cr1 | cr2 | cr3 | cr4 | cr5 | cr6 | cr7 | cr8 | ca1 | ca2 | ca3 | ca4 | rs1 | rs2 | ps1 | ps2 | ps3 | ps4 |
|---|---|---|---|---|---|---|---|---|---|---|---|---|---|---|---|---|---|---|
| cr1 | 1 | 0.77 | 0.6 | 0.59 | 0.55 | 0.5 | 0.31 | 0.32 | 0.68 | 0.53 | 0.26 | 0.16 | 0.31 | 0.11 | 0 | −0.08 | −0.12 | −0.08 |
| cr2 | 0.77 | 1 | 0.49 | 0.63 | 0.47 | 0.48 | 0.25 | 0.34 | 0.64 | 0.44 | 0.18 | 0.09 | 0.19 | 0.04 | 0.07 | −0.05 | −0.03 | 0.02 |
| cr3 | 0.6 | 0.49 | 1 | 0.73 | 0.47 | 0.46 | 0.3 | 0.28 | 0.51 | 0.67 | 0.28 | 0.04 | 0.29 | 0.16 | −0.04 | −0.07 | −0.06 | −0.16 |
| cr4 | 0.59 | 0.63 | 0.73 | 1 | 0.51 | 0.49 | 0.35 | 0.32 | 0.43 | 0.67 | 0.16 | 0.02 | 0.25 | 0.08 | −0.05 | −0.06 | −0.06 | −0.16 |
| cr5 | 0.55 | 0.47 | 0.47 | 0.51 | 1 | 0.82 | 0.36 | 0.39 | 0.45 | 0.41 | 0.5 | 0.27 | 0.12 | 0.08 | −0.04 | −0.1 | −0.02 | 0.14 |
| cr6 | 0.5 | 0.48 | 0.46 | 0.49 | 0.82 | 1 | 0.44 | 0.45 | 0.43 | 0.39 | 0.57 | 0.3 | 0.12 | 0.14 | −0.02 | −0.07 | −0.11 | 0.11 |
| cr7 | 0.31 | 0.25 | 0.3 | 0.35 | 0.36 | 0.44 | 1 | 0.87 | 0.3 | 0.21 | 0.38 | 0.57 | 0.12 | 0.23 | 0.02 | 0.02 | −0.12 | 0.12 |
| cr8 | 0.32 | 0.34 | 0.28 | 0.32 | 0.39 | 0.45 | 0.87 | 1 | 0.3 | 0.2 | 0.35 | 0.55 | 0.12 | 0.11 | −0.01 | −0.01 | −0.12 | 0.06 |
| ca1 | 0.68 | 0.64 | 0.51 | 0.43 | 0.45 | 0.43 | 0.3 | 0.3 | 1 | 0.59 | 0.52 | 0.39 | 0.35 | 0.34 | 0.1 | −0.02 | −0.03 | 0.14 |
| ca2 | 0.53 | 0.44 | 0.67 | 0.67 | 0.41 | 0.39 | 0.21 | 0.2 | 0.59 | 1 | 0.46 | 0.15 | 0.52 | 0.23 | 0.08 | −0.03 | −0.02 | −0.01 |
| ca3 | 0.26 | 0.18 | 0.28 | 0.16 | 0.5 | 0.57 | 0.38 | 0.35 | 0.52 | 0.46 | 1 | 0.57 | 0.22 | 0.32 | 0.22 | 0.2 | 0.19 | 0.35 |
| ca4 | 0.16 | 0.09 | 0.04 | 0.02 | 0.27 | 0.3 | 0.57 | 0.55 | 0.39 | 0.15 | 0.57 | 1 | 0.22 | 0.35 | −0.08 | 0.01 | 0.1 | 0.15 |
| rs1 | 0.31 | 0.19 | 0.29 | 0.25 | 0.12 | 0.12 | 0.12 | 0.12 | 0.35 | 0.52 | 0.22 | 0.22 | 1 | 0.31 | 0 | 0 | −0.13 | −0.07 |
| rs2 | 0.11 | 0.04 | 0.16 | 0.08 | 0.08 | 0.14 | 0.23 | 0.11 | 0.34 | 0.23 | 0.32 | 0.35 | 0.31 | 1 | −0.01 | 0.14 | −0.08 | 0.13 |
| ps1 | 0 | 0.07 | −0.04 | −0.05 | −0.04 | −0.02 | 0.02 | −0.01 | 0.1 | 0.08 | 0.22 | −0.08 | 0 | −0.01 | 1 | 0.5 | 0.33 | 0.32 |
| ps2 | −0.08 | −0.05 | −0.07 | −0.06 | −0.1 | −0.07 | 0.02 | −0.01 | −0.02 | −0.03 | 0.2 | 0.01 | 0 | 0.14 | 0.5 | 1 | 0.31 | 0.11 |
| ps3 | −0.12 | −0.03 | −0.06 | −0.06 | −0.02 | −0.11 | −0.12 | −0.12 | −0.03 | −0.02 | 0.19 | 0.1 | −0.13 | −0.08 | 0.33 | 0.31 | 1 | 0.5 |
| ps4 | −0.08 | 0.02 | −0.16 | −0.16 | 0.14 | 0.11 | 0.12 | 0.06 | 0.14 | −0.01 | 0.35 | 0.15 | −0.07 | 0.13 | 0.32 | 0.11 | 0.5 | 1 |
| pv1 | −0.24 | −0.17 | −0.32 | −0.24 | −0.26 | −0.23 | −0.22 | −0.19 | −0.28 | −0.35 | −0.32 | −0.24 | −0.31 | −0.18 | 0.24 | 0.14 | 0.13 | 0.02 |
| pv2 | 0 | 0.02 | −0.12 | −0.02 | −0.1 | −0.12 | −0.19 | −0.15 | −0.26 | −0.28 | −0.29 | −0.3 | −0.36 | −0.22 | 0.09 | 0.09 | 0 | −0.09 |
| pv3 | −0.16 | 0 | −0.15 | −0.12 | −0.14 | −0.14 | −0.2 | −0.14 | −0.11 | −0.26 | −0.22 | −0.21 | −0.39 | −0.22 | 0.09 | 0.12 | 0.21 | 0.18 |
| pv4 | −0.31 | −0.21 | −0.17 | −0.21 | −0.1 | −0.11 | 0.02 | −0.05 | −0.13 | −0.2 | −0.07 | −0.19 | −0.35 | −0.18 | 0.22 | 0.19 | 0.17 | 0.37 |
| rc1 | −0.27 | −0.2 | −0.22 | −0.2 | −0.15 | −0.1 | −0.03 | −0.02 | −0.19 | −0.21 | −0.05 | 0.02 | −0.11 | 0.06 | −0.09 | 0.01 | −0.02 | 0.02 |
| rc2 | −0.1 | −0.08 | −0.25 | −0.28 | −0.03 | 0.04 | −0.16 | −0.15 | −0.09 | −0.28 | −0.01 | 0.05 | −0.19 | 0.07 | −0.03 | 0 | 0 | −0.03 |
| rc3 | −0.14 | −0.04 | −0.1 | −0.14 | −0.16 | −0.25 | −0.05 | 0.02 | −0.12 | −0.26 | −0.3 | −0.05 | −0.2 | −0.07 | −0.02 | 0.09 | 0.03 | −0.14 |
| rc4 | 0 | −0.1 | 0.05 | −0.02 | 0.02 | −0.03 | −0.19 | −0.24 | 0.14 | 0 | 0.09 | −0.1 | −0.12 | 0.1 | 0.03 | 0.09 | 0.07 | 0.06 |
| crd1 | 0.04 | 0.04 | −0.04 | −0.11 | −0.04 | 0.03 | −0.02 | 0.07 | 0.14 | −0.05 | 0.26 | 0.28 | −0.05 | 0.05 | 0.21 | 0.12 | 0.12 | 0.1 |
| crd2 | 0.08 | 0.08 | 0.05 | −0.03 | 0.1 | 0.09 | 0.08 | 0.14 | 0.24 | 0.11 | 0.41 | 0.35 | −0.05 | 0.17 | 0.15 | 0.1 | 0.18 | 0.22 |
| re1 | 0.44 | 0.41 | 0.31 | 0.39 | 0.3 | 0.29 | 0.29 | 0.3 | 0.31 | 0.49 | 0.3 | 0.29 | 0.39 | 0.19 | 0.04 | 0 | 0.11 | −0.05 |
| re2 | 0.21 | 0.21 | 0.35 | 0.32 | 0.23 | 0.11 | 0.28 | 0.24 | 0.33 | 0.5 | 0.27 | 0.16 | 0.39 | 0.17 | 0.09 | 0 | −0.07 | 0.01 |
| re3 | 0.16 | 0.19 | 0.15 | 0.12 | 0.35 | 0.3 | 0.16 | 0.15 | 0.26 | 0.34 | 0.39 | 0.32 | 0.17 | 0.15 | 0.08 | 0.07 | 0.15 | 0.31 |
| re4 | 0.05 | 0.05 | 0.14 | 0.2 | 0.16 | 0.12 | 0.12 | 0.11 | 0.09 | 0.23 | 0.21 | 0.13 | −0.04 | 0.14 | −0.05 | −0.06 | 0.08 | 0 |
| rfs1 | −0.11 | −0.18 | −0.06 | −0.16 | −0.01 | −0.06 | 0.02 | 0.03 | −0.13 | −0.03 | 0.02 | 0.1 | 0 | 0.08 | −0.13 | 0 | 0.11 | 0.13 |
| rfs2 | 0.1 | −0.1 | −0.06 | −0.11 | −0.02 | −0.08 | −0.03 | −0.08 | −0.01 | −0.08 | −0.06 | 0.09 | 0.02 | −0.01 | −0.3 | −0.09 | −0.17 | −0.14 |
| rfs3 | −0.03 | 0.08 | 0.16 | 0.11 | −0.02 | −0.07 | −0.1 | −0.04 | 0 | 0.04 | 0.09 | −0.03 | 0.02 | 0.1 | −0.05 | 0.05 | 0.06 | −0.06 |
| rfs4 | 0.08 | −0.13 | −0.08 | −0.13 | 0.07 | 0.02 | 0.11 | 0.01 | −0.03 | −0.1 | 0.13 | 0.21 | −0.02 | 0.17 | −0.15 | 0.03 | −0.07 | 0.02 |
| se1 | 0.6 | 0.51 | 0.47 | 0.36 | 0.34 | 0.37 | 0.26 | 0.27 | 0.61 | 0.43 | 0.33 | 0.38 | 0.25 | 0.28 | 0.07 | 0.06 | 0.05 | −0.08 |
| se2 | 0.41 | 0.45 | 0.21 | 0.27 | 0.32 | 0.29 | 0.13 | 0.17 | 0.45 | 0.15 | 0.08 | 0.23 | 0.08 | 0.2 | 0.09 | 0 | 0.11 | 0 |
| se3 | 0.42 | 0.34 | 0.47 | 0.45 | 0.37 | 0.33 | 0.12 | 0.13 | 0.48 | 0.64 | 0.37 | 0.13 | 0.4 | 0.18 | 0.11 | 0.07 | −0.05 | −0.05 |
| se4 | 0.37 | 0.38 | 0.46 | 0.59 | 0.32 | 0.23 | 0.32 | 0.3 | 0.3 | 0.42 | 0.13 | 0.08 | 0.19 | 0.05 | −0.09 | 0.05 | 0.08 | −0.06 |
| se5 | 0.21 | 0.1 | 0.22 | 0.08 | 0.34 | 0.51 | 0.41 | 0.36 | 0.31 | 0.24 | 0.59 | 0.39 | 0.11 | 0.27 | −0.04 | −0.05 | −0.02 | 0.16 |
| se6 | 0.3 | 0.27 | 0.35 | 0.27 | 0.47 | 0.55 | 0.19 | 0.16 | 0.29 | 0.27 | 0.43 | 0.1 | −0.12 | 0.1 | −0.01 | −0.14 | 0.04 | 0.21 |
| se7 | 0.24 | 0.15 | 0.17 | 0.13 | 0.27 | 0.3 | 0.6 | 0.61 | 0.31 | 0.17 | 0.44 | 0.73 | 0.1 | 0.22 | −0.07 | −0.04 | 0.09 | 0.11 |
| se8 | 0.15 | 0.21 | 0.07 | 0.14 | 0.22 | 0.19 | 0.44 | 0.53 | 0.08 | 0.02 | 0.2 | 0.48 | −0.08 | 0.01 | −0.02 | −0.02 | 0.09 | 0.17 |
| si1 | 0.22 | 0.22 | 0.17 | 0.17 | 0.16 | 0.2 | 0.12 | 0.12 | 0.27 | 0.26 | 0.25 | 0.25 | 0.09 | 0.15 | 0.23 | 0.1 | 0.14 | −0.03 |
| si2 | 0.14 | 0.14 | 0.25 | 0.25 | 0.29 | 0.24 | 0.21 | 0.23 | 0.18 | 0.3 | 0.2 | 0.25 | 0.06 | 0.08 | 0.04 | −0.01 | −0.04 | |
| si3 | 0.12 | 0.02 | 0.06 | −0.05 | 0.23 | 0.29 | 0.2 | 0.16 | 0.17 | 0.25 | 0.56 | 0.39 | 0.16 | 0.16 | 0.12 | 0.03 | 0.29 | 0.31 |
| si4 | −0.05 | 0.02 | −0.03 | 0.03 | 0.13 | 0.15 | 0.31 | 0.32 | 0.02 | 0.04 | 0.33 | 0.34 | −0.03 | 0.06 | 0.04 | 0.03 | 0.19 | 0.14 |
| sn1 | 0 | 0.04 | 0.09 | 0.01 | 0.16 | 0.19 | 0.15 | 0.1 | 0.19 | 0.2 | 0.35 | 0.21 | 0.09 | 0.14 | 0.04 | 0.01 | 0.12 | 0.21 |
| sn2 | −0.05 | −0.07 | 0.15 | 0.11 | 0.17 | 0.14 | 0.05 | 0.01 | 0.07 | 0.2 | 0.29 | 0.1 | 0.06 | 0.05 | −0.05 | 0.17 | 0.13 | 0.09 |
| sn3 | −0.07 | 0.02 | −0.01 | −0.12 | 0.05 | 0.13 | 0.11 | 0.14 | 0.06 | 0.13 | 0.33 | 0.19 | −0.02 | 0 | 0.05 | −0.01 | 0.07 | 0.24 |
| sn4 | −0.21 | −0.1 | −0.22 | −0.24 | −0.13 | −0.09 | −0.03 | −0.03 | −0.06 | −0.06 | 0.24 | 0.13 | −0.12 | −0.12 | 0.11 | 0.19 | 0.36 | 0.28 |
| ta1 | 0.2 | 0.12 | 0.22 | 0.09 | 0.05 | 0.17 | 0.24 | 0.23 | 0.41 | 0.21 | 0.44 | 0.41 | 0.06 | 0.25 | 0.21 | 0.09 | 0.02 | 0.06 |
| ta2 | 0.17 | 0.07 | 0.11 | 0.03 | 0.11 | 0.12 | 0.19 | 0.17 | 0.34 | 0.23 | 0.46 | 0.42 | 0.25 | 0.31 | 0.22 | 0.32 | −0.01 | 0.04 |
| ta3 | 0.07 | −0.01 | 0.06 | 0.06 | 0.14 | 0.17 | 0.36 | 0.31 | 0.28 | 0.16 | 0.43 | 0.42 | 0.12 | 0.33 | 0.18 | 0.22 | 0.07 | 0.29 |
| ta4 | 0.15 | 0.01 | 0.16 | 0.09 | 0.12 | 0.2 | 0.31 | 0.21 | 0.29 | 0.2 | 0.4 | 0.42 | 0.14 | 0.29 | 0.09 | 0.06 | −0.02 | 0.18 |

**Table A2.** *Cont.*

| | pv1 | pv2 | pv3 | pv4 | rc1 | rc2 | rc3 | rc4 | crd1 | crd2 | re1 | re2 | re3 | re4 | rfs1 | rfs2 | rfs3 | rfs4 |
|---|---|---|---|---|---|---|---|---|---|---|---|---|---|---|---|---|---|---|
| cr1 | −0.24 | 0 | −0.16 | −0.31 | −0.27 | −0.1 | −0.14 | 0 | 0.04 | 0.08 | 0.44 | 0.21 | 0.16 | 0.05 | −0.11 | 0.1 | −0.03 | 0.08 |
| cr2 | −0.17 | 0.02 | 0 | −0.21 | −0.2 | −0.08 | −0.04 | −0.1 | 0.04 | 0.08 | 0.41 | 0.21 | 0.19 | 0.05 | −0.18 | −0.1 | 0.08 | −0.13 |
| cr3 | −0.32 | −0.12 | −0.15 | −0.17 | −0.22 | −0.25 | −0.1 | 0.05 | −0.04 | 0.05 | 0.31 | 0.35 | 0.15 | 0.14 | −0.06 | −0.06 | 0.16 | −0.08 |
| cr4 | −0.24 | −0.02 | −0.12 | −0.21 | −0.2 | −0.28 | −0.14 | −0.02 | −0.11 | −0.03 | 0.39 | 0.32 | 0.12 | 0.2 | −0.16 | −0.11 | 0.11 | −0.13 |
| cr5 | −0.26 | −0.1 | −0.14 | −0.1 | −0.15 | −0.03 | −0.16 | 0.02 | −0.04 | 0.1 | 0.3 | 0.23 | 0.35 | 0.16 | −0.01 | −0.02 | −0.02 | 0.07 |
| cr6 | −0.23 | −0.12 | −0.14 | −0.11 | −0.1 | 0.04 | −0.25 | −0.03 | 0.03 | 0.09 | 0.29 | 0.11 | 0.3 | 0.12 | −0.06 | −0.08 | −0.07 | 0.02 |
| cr7 | −0.22 | −0.19 | −0.2 | 0.02 | −0.03 | −0.16 | −0.05 | −0.19 | −0.02 | 0.08 | 0.29 | 0.28 | 0.16 | 0.12 | 0.02 | −0.03 | −0.1 | 0.11 |
| cr8 | −0.19 | −0.15 | −0.14 | −0.05 | −0.02 | −0.15 | 0.02 | −0.24 | 0.07 | 0.14 | 0.3 | 0.24 | 0.15 | 0.11 | 0.03 | −0.08 | −0.04 | 0.01 |
| ca1 | −0.28 | −0.26 | −0.11 | −0.13 | −0.19 | −0.09 | −0.12 | 0.14 | 0.14 | 0.24 | 0.31 | 0.33 | 0.26 | 0.09 | −0.13 | −0.01 | 0 | −0.03 |
| ca2 | −0.35 | −0.28 | −0.26 | −0.2 | −0.21 | −0.28 | −0.26 | 0 | −0.05 | 0.11 | 0.49 | 0.5 | 0.34 | 0.23 | −0.03 | −0.08 | 0.04 | −0.1 |
| ca3 | −0.32 | −0.29 | −0.22 | −0.07 | −0.05 | −0.01 | −0.3 | 0.09 | 0.26 | 0.41 | 0.3 | 0.27 | 0.39 | 0.21 | 0.02 | −0.06 | 0.09 | 0.13 |
| ca4 | −0.24 | −0.3 | −0.21 | −0.19 | 0.02 | 0.05 | −0.05 | −0.1 | 0.28 | 0.35 | 0.29 | 0.16 | 0.32 | 0.13 | 0.1 | 0.09 | −0.03 | 0.21 |
| rs1 | −0.31 | −0.36 | −0.39 | −0.35 | −0.11 | −0.19 | −0.2 | −0.12 | −0.05 | −0.05 | 0.39 | 0.39 | 0.17 | −0.04 | 0 | 0.02 | 0.02 | −0.02 |
| rs2 | −0.18 | −0.22 | −0.22 | −0.18 | 0.06 | 0.07 | −0.07 | 0.1 | 0.05 | 0.17 | 0.19 | 0.17 | 0.15 | 0.14 | 0.08 | −0.01 | 0.1 | 0.17 |
| ps1 | 0.24 | 0.09 | 0.09 | 0.22 | −0.09 | −0.03 | −0.02 | 0.03 | 0.21 | 0.15 | 0.04 | 0.09 | 0.08 | −0.05 | −0.13 | −0.3 | −0.05 | −0.15 |
| ps2 | 0.14 | 0.09 | 0.12 | 0.19 | 0.01 | 0 | 0.09 | 0.09 | 0.12 | 0.1 | 0 | 0 | 0.07 | −0.06 | 0 | −0.09 | 0.05 | 0.03 |
| ps3 | 0.13 | 0 | 0.21 | 0.17 | −0.02 | 0 | 0.03 | 0.07 | 0.12 | 0.18 | 0.11 | −0.07 | 0.15 | 0.08 | 0.11 | −0.17 | 0.06 | −0.07 |
| ps4 | 0.02 | −0.09 | 0.18 | 0.37 | 0.02 | −0.03 | −0.14 | 0.06 | 0.1 | 0.22 | −0.05 | 0.01 | 0.31 | 0 | 0.13 | −0.14 | −0.06 | 0.02 |
| pv1 | 1 | 0.62 | 0.42 | 0.39 | 0.04 | 0.17 | 0.07 | 0.01 | 0.14 | 0.04 | −0.15 | −0.29 | −0.15 | −0.01 | 0.04 | 0.04 | 0.02 | −0.24 |
| pv2 | 0.62 | 1 | 0.24 | 0.13 | −0.15 | 0.09 | 0.02 | −0.12 | 0.14 | 0.01 | −0.12 | −0.31 | −0.23 | −0.19 | −0.11 | 0.16 | −0.01 | −0.18 |
| pv3 | 0.42 | 0.24 | 1 | 0.48 | 0.08 | 0.08 | 0.06 | 0.13 | 0.06 | −0.03 | −0.14 | −0.22 | −0.15 | 0 | −0.04 | 0 | 0.03 | −0.09 |
| pv4 | 0.39 | 0.13 | 0.48 | 1 | 0.04 | 0.02 | 0.04 | 0.14 | 0.07 | 0.05 | −0.23 | −0.04 | 0.06 | −0.08 | 0.1 | −0.04 | −0.15 | −0.11 |
| rc1 | 0.04 | −0.15 | 0.08 | 0.04 | 1 | 0.45 | 0.39 | 0.31 | −0.08 | −0.11 | −0.12 | −0.09 | 0.1 | 0.01 | −0.05 | 0.01 | 0.08 | −0.02 |
| rc2 | 0.17 | 0.09 | 0.08 | 0.02 | 0.45 | 1 | 0.4 | 0.17 | 0.17 | 0.07 | −0.03 | −0.16 | 0.18 | 0.04 | −0.12 | 0.1 | −0.07 | 0.05 |
| rc3 | 0.07 | 0.02 | 0.06 | 0.04 | 0.39 | 0.4 | 1 | 0.22 | 0.02 | −0.06 | −0.23 | −0.15 | −0.13 | 0 | −0.02 | 0.06 | 0.02 | 0.15 |
| rc4 | 0.01 | −0.12 | 0.13 | 0.14 | 0.31 | 0.17 | 0.22 | 1 | 0.09 | −0.05 | −0.24 | −0.14 | −0.04 | 0.09 | −0.02 | 0.02 | 0.08 | 0.04 |
| crd1 | 0.14 | 0.14 | 0.06 | 0.07 | −0.08 | 0.17 | 0.02 | 0.09 | 1 | 0.74 | −0.08 | −0.11 | 0.02 | 0 | −0.02 | 0.04 | 0.03 | 0.07 |
| crd2 | 0.04 | 0.01 | −0.03 | 0.05 | −0.11 | 0.07 | −0.06 | −0.05 | 0.74 | 1 | 0.08 | 0.04 | 0.09 | 0.13 | 0.01 | −0.02 | 0.09 | 0.11 |
| re1 | −0.15 | −0.12 | −0.14 | −0.23 | −0.12 | −0.03 | −0.23 | −0.24 | −0.08 | 0.08 | 1 | 0.62 | 0.48 | 0.27 | 0.13 | 0.1 | 0.07 | −0.18 |
| re2 | −0.29 | −0.31 | −0.22 | −0.04 | −0.09 | −0.16 | −0.15 | −0.14 | −0.11 | 0.04 | 0.62 | 1 | 0.48 | 0.21 | 0.11 | −0.07 | 0.07 | −0.18 |
| re3 | −0.15 | −0.23 | −0.15 | 0.06 | 0.1 | 0.18 | −0.13 | −0.04 | 0.02 | 0.09 | 0.48 | 0.48 | 1 | 0.34 | 0.27 | −0.01 | 0 | −0.04 |
| re4 | −0.01 | −0.19 | 0 | −0.08 | 0.01 | 0.04 | 0 | 0.09 | 0 | 0.13 | 0.27 | 0.21 | 0.34 | 1 | 0.23 | 0.02 | 0.28 | 0.05 |
| rfs1 | 0.04 | −0.11 | −0.04 | 0.1 | −0.05 | −0.12 | −0.02 | −0.02 | −0.02 | 0.01 | 0.13 | 0.11 | 0.27 | 0.23 | 1 | 0.18 | 0.11 | 0.02 |
| rfs2 | 0.04 | 0.16 | 0 | −0.04 | 0.01 | 0.1 | 0.06 | 0.02 | 0.04 | −0.02 | 0.1 | −0.07 | −0.01 | 0.02 | 0.18 | 1 | 0.16 | 0.07 |
| rfs3 | 0.02 | −0.01 | 0.03 | −0.15 | 0.08 | −0.07 | 0.02 | 0.08 | 0.03 | 0.09 | 0.07 | 0.07 | 0 | 0.28 | 0.11 | 0.16 | 1 | 0 |
| rfs4 | −0.24 | −0.18 | −0.09 | −0.11 | −0.02 | 0.05 | 0.15 | 0.04 | 0.07 | 0.11 | −0.18 | −0.18 | −0.04 | 0.05 | 0.02 | 0.07 | 0 | 1 |
| se1 | −0.11 | −0.03 | −0.09 | −0.24 | −0.31 | −0.04 | −0.08 | −0.04 | 0.21 | 0.29 | 0.47 | 0.18 | 0.21 | 0.09 | 0.03 | 0.16 | 0.05 | 0 |
| se2 | 0.01 | 0.02 | 0.05 | −0.13 | −0.44 | −0.04 | −0.02 | −0.03 | 0.03 | 0.16 | 0.27 | 0.1 | 0.08 | 0.13 | −0.01 | −0.03 | −0.02 | 0.08 |
| se3 | −0.15 | −0.16 | −0.22 | −0.09 | −0.23 | −0.37 | −0.17 | 0.02 | 0.06 | 0.21 | 0.31 | 0.43 | 0.18 | 0.15 | 0.03 | −0.05 | 0.08 | −0.13 |
| se4 | −0.31 | −0.1 | −0.13 | −0.1 | −0.27 | −0.57 | −0.08 | −0.02 | −0.19 | −0.06 | 0.24 | 0.35 | 0 | 0.07 | 0.05 | −0.02 | 0 | −0.09 |
| se5 | −0.15 | −0.1 | −0.12 | 0.01 | −0.09 | 0 | −0.33 | −0.11 | 0.22 | 0.29 | 0.22 | 0.17 | 0.22 | 0.17 | 0.14 | 0.03 | −0.02 | 0.04 |
| se6 | −0.22 | −0.19 | 0.03 | 0.07 | −0.08 | −0.02 | −0.34 | −0.06 | 0 | 0.16 | 0.13 | 0.13 | 0.29 | 0.22 | 0.08 | 0 | 0.04 | −0.01 |
| se7 | −0.17 | −0.22 | −0.11 | −0.18 | 0.09 | 0.09 | 0.05 | −0.26 | 0.18 | 0.32 | 0.31 | 0.14 | 0.27 | 0.31 | 0.14 | 0.13 | 0.06 | 0.2 |
| se8 | −0.14 | −0.16 | −0.04 | −0.11 | 0.02 | 0.09 | 0.11 | −0.4 | 0 | 0.17 | 0.27 | 0.2 | 0.3 | 0.29 | 0.08 | −0.04 | 0.07 | 0.12 |
| si1 | 0.1 | 0.01 | −0.07 | −0.1 | −0.37 | −0.07 | −0.12 | −0.08 | 0.19 | 0.37 | 0.37 | 0.09 | 0.05 | 0.17 | 0.12 | 0.02 | 0.1 | −0.07 |
| si2 | −0.05 | −0.17 | −0.18 | −0.03 | −0.32 | −0.35 | −0.13 | −0.14 | 0.05 | 0.21 | 0.22 | 0.33 | 0.12 | 0.14 | 0.26 | −0.04 | 0.17 | −0.11 |
| si3 | −0.14 | −0.14 | −0.14 | −0.03 | −0.12 | −0.02 | −0.02 | −0.36 | 0.27 | 0.4 | 0.31 | 0.21 | 0.37 | 0.13 | 0.26 | 0.03 | −0.06 | 0.01 |
| si4 | 0.1 | 0.01 | −0.04 | 0.02 | −0.14 | 0.07 | −0.01 | −0.36 | 0.21 | 0.38 | 0.29 | 0.11 | 0.16 | 0.32 | 0.14 | −0.02 | 0.13 | 0.02 |
| sn1 | −0.22 | −0.3 | −0.08 | 0 | −0.21 | −0.05 | −0.17 | 0 | 0.12 | 0.25 | 0.24 | 0.2 | 0.28 | 0.21 | 0.3 | 0.01 | 0.16 | 0.02 |
| sn2 | −0.29 | −0.29 | −0.2 | −0.05 | −0.12 | −0.23 | −0.06 | 0.09 | −0.04 | 0.15 | 0.04 | 0.17 | 0.18 | 0.15 | 0.25 | −0.06 | 0.23 | 0.09 |
| sn3 | −0.12 | −0.23 | −0.1 | 0 | −0.08 | 0.03 | −0.07 | −0.01 | 0.24 | 0.28 | 0.07 | 0.08 | 0.26 | 0.23 | 0.37 | −0.05 | 0.13 | 0.03 |
| sn4 | 0.03 | −0.13 | 0.08 | 0.17 | −0.06 | 0.03 | −0.01 | −0.02 | 0.14 | 0.26 | 0.05 | 0.01 | 0.22 | 0.24 | 0.26 | −0.08 | 0.17 | 0.04 |
| ta1 | 0.03 | 0.04 | −0.08 | 0.01 | −0.09 | 0.07 | −0.11 | 0.05 | 0.45 | 0.37 | 0.12 | 0.05 | 0.08 | 0.02 | −0.24 | 0.1 | 0.11 | −0.04 |
| ta2 | −0.04 | −0.08 | −0.22 | −0.08 | −0.03 | 0.03 | −0.04 | 0.11 | 0.35 | 0.33 | 0.09 | 0.11 | 0.14 | 0.03 | −0.13 | 0.08 | 0.01 | 0.06 |
| ta3 | −0.04 | −0.11 | −0.27 | 0.09 | −0.02 | 0.01 | −0.02 | −0.01 | 0.38 | 0.35 | 0.05 | 0.05 | 0.27 | 0.01 | 0.09 | 0.14 | 0.04 | 0.11 |
| ta4 | −0.05 | −0.04 | −0.2 | 0.05 | −0.03 | 0.03 | −0.01 | −0.03 | 0.28 | 0.29 | 0.1 | 0.07 | 0.2 | −0.05 | 0.04 | 0.17 | 0.07 | 0.25 |

**Table A2.** *Cont.*

| | se1 | se2 | se3 | se4 | se5 | se6 | se7 | se8 | si1 | si2 | si3 | si4 | sn1 | sn2 | sn3 | sn4 | ta1 | ta2 | ta3 | ta4 |
|---|---|---|---|---|---|---|---|---|---|---|---|---|---|---|---|---|---|---|---|---|
| cr1 | 0.6 | 0.41 | 0.42 | 0.37 | 0.21 | 0.3 | 0.24 | 0.15 | 0.22 | 0.14 | 0.12 | −0.05 | 0 | −0.05 | −0.07 | −0.21 | 0.2 | 0.17 | 0.07 | 0.15 |
| cr2 | 0.51 | 0.45 | 0.34 | 0.38 | 0.1 | 0.27 | 0.15 | 0.21 | 0.22 | 0.14 | 0.02 | 0.02 | 0.04 | −0.07 | 0.02 | −0.1 | 0.12 | 0.07 | −0.01 | 0.01 |
| cr3 | 0.47 | 0.21 | 0.47 | 0.46 | 0.22 | 0.35 | 0.17 | 0.07 | 0.17 | 0.25 | 0.06 | −0.03 | 0.09 | 0.15 | −0.01 | −0.22 | 0.22 | 0.11 | 0.06 | 0.16 |
| cr4 | 0.36 | 0.27 | 0.45 | 0.59 | 0.08 | 0.27 | 0.13 | 0.14 | 0.17 | 0.25 | −0.05 | 0.03 | 0.01 | 0.11 | −0.12 | −0.24 | 0.09 | 0.03 | 0.06 | 0.09 |
| cr5 | 0.34 | 0.32 | 0.37 | 0.32 | 0.34 | 0.47 | 0.27 | 0.22 | 0.16 | 0.29 | 0.23 | 0.13 | 0.16 | 0.17 | 0.05 | −0.13 | 0.05 | 0.11 | 0.14 | 0.12 |
| cr6 | 0.37 | 0.29 | 0.33 | 0.23 | 0.51 | 0.55 | 0.3 | 0.19 | 0.2 | 0.24 | 0.29 | 0.15 | 0.19 | 0.14 | 0.13 | −0.09 | 0.17 | 0.12 | 0.17 | 0.2 |
| cr7 | 0.26 | 0.13 | 0.12 | 0.32 | 0.41 | 0.19 | 0.6 | 0.44 | 0.12 | 0.21 | 0.2 | 0.31 | 0.15 | 0.05 | 0.11 | −0.03 | 0.24 | 0.19 | 0.36 | 0.31 |
| cr8 | 0.27 | 0.17 | 0.13 | 0.3 | 0.36 | 0.16 | 0.61 | 0.53 | 0.12 | 0.23 | 0.16 | 0.32 | 0.1 | 0.01 | 0.14 | −0.03 | 0.23 | 0.17 | 0.31 | 0.21 |
| ca1 | 0.61 | 0.45 | 0.48 | 0.3 | 0.31 | 0.29 | 0.31 | 0.08 | 0.27 | 0.18 | 0.17 | 0.02 | 0.19 | 0.07 | 0.06 | −0.06 | 0.41 | 0.34 | 0.28 | 0.29 |
| ca2 | 0.43 | 0.15 | 0.64 | 0.42 | 0.24 | 0.27 | 0.17 | 0.02 | 0.26 | 0.3 | 0.25 | 0.04 | 0.2 | 0.2 | 0.13 | −0.03 | 0.21 | 0.23 | 0.16 | 0.2 |
| ca3 | 0.33 | 0.08 | 0.37 | 0.13 | 0.59 | 0.43 | 0.44 | 0.2 | 0.25 | 0.25 | 0.56 | 0.33 | 0.35 | 0.29 | 0.33 | 0.24 | 0.44 | 0.46 | 0.43 | 0.4 |
| ca4 | 0.38 | 0.23 | 0.13 | 0.08 | 0.39 | 0.1 | 0.73 | 0.48 | 0.25 | 0.2 | 0.39 | 0.34 | 0.21 | 0.1 | 0.19 | 0.13 | 0.41 | 0.42 | 0.42 | 0.42 |
| rs1 | 0.25 | 0.08 | 0.4 | 0.19 | 0.11 | −0.12 | 0.1 | −0.08 | 0.09 | 0.23 | 0.16 | −0.03 | 0.09 | 0.06 | −0.02 | −0.12 | 0.06 | 0.25 | 0.12 | 0.14 |
| rs2 | 0.28 | 0.2 | 0.18 | 0.05 | 0.27 | 0.1 | 0.22 | 0.01 | 0.15 | 0.06 | 0.16 | 0.06 | 0.14 | 0.05 | 0 | −0.12 | 0.25 | 0.31 | 0.33 | 0.29 |
| ps1 | 0.07 | 0.09 | 0.11 | −0.09 | −0.04 | −0.01 | −0.07 | −0.02 | 0.23 | 0.08 | 0.12 | 0.04 | 0.04 | −0.05 | 0.05 | 0.11 | 0.21 | 0.22 | 0.18 | 0.09 |
| ps2 | 0.06 | 0 | 0.07 | 0.05 | −0.05 | −0.14 | −0.04 | −0.02 | 0.1 | 0.04 | 0.03 | 0.03 | 0.01 | 0.17 | −0.01 | 0.19 | 0.09 | 0.32 | 0.22 | 0.06 |
| ps3 | 0.05 | 0.11 | −0.05 | 0.08 | −0.02 | 0.04 | 0.09 | 0.09 | 0.14 | −0.01 | 0.29 | 0.19 | 0.12 | 0.13 | 0.07 | 0.36 | 0.02 | −0.01 | 0.07 | −0.02 |
| ps4 | −0.08 | 0 | −0.05 | −0.06 | 0.16 | 0.21 | 0.11 | 0.17 | −0.03 | −0.04 | 0.31 | 0.14 | 0.21 | 0.09 | 0.24 | 0.28 | 0.06 | 0.04 | 0.29 | 0.18 |
| pv1 | −0.11 | 0.01 | −0.15 | −0.31 | −0.15 | −0.22 | −0.17 | −0.14 | −0.1 | −0.05 | −0.14 | 0.1 | −0.22 | −0.29 | −0.12 | 0.03 | 0.03 | −0.04 | −0.04 | −0.05 |
| pv2 | −0.03 | 0.02 | −0.16 | −0.1 | −0.1 | −0.19 | −0.22 | −0.16 | 0.01 | −0.17 | −0.14 | 0.01 | −0.3 | −0.29 | −0.23 | −0.13 | 0.04 | −0.08 | −0.11 | −0.04 |
| pv3 | −0.09 | 0.05 | −0.22 | −0.13 | −0.12 | 0.03 | −0.11 | −0.04 | −0.07 | −0.18 | −0.14 | −0.04 | −0.08 | −0.2 | −0.1 | 0.08 | −0.08 | −0.22 | −0.27 | −0.2 |
| pv4 | −0.24 | −0.13 | −0.09 | 0.1 | 0.01 | 0.07 | −0.18 | −0.11 | −0.1 | −0.03 | −0.03 | 0.02 | 0 | −0.05 | 0 | 0.17 | 0.01 | −0.08 | 0.09 | 0.05 |
| rc1 | −0.31 | −0.44 | −0.23 | −0.27 | −0.09 | −0.08 | 0.09 | 0.02 | −0.37 | −0.32 | −0.12 | −0.14 | −0.21 | −0.12 | −0.08 | −0.06 | −0.09 | −0.03 | −0.02 | −0.03 |
| rc2 | −0.04 | −0.04 | −0.37 | −0.57 | 0 | −0.02 | 0.09 | 0.09 | −0.07 | −0.35 | −0.02 | 0.07 | −0.05 | −0.23 | 0.03 | 0.03 | 0.07 | 0.03 | 0.01 | 0.03 |
| rc3 | −0.08 | −0.02 | −0.17 | −0.08 | −0.33 | −0.34 | 0.05 | 0.11 | −0.12 | −0.13 | −0.36 | −0.01 | −0.17 | −0.06 | −0.07 | −0.01 | −0.11 | −0.04 | −0.02 | −0.01 |
| rc4 | −0.04 | −0.03 | 0.02 | −0.02 | −0.11 | −0.06 | −0.26 | −0.4 | −0.08 | −0.14 | −0.21 | −0.36 | 0 | 0.09 | −0.01 | −0.02 | 0.05 | 0.11 | −0.01 | −0.03 |
| crd1 | 0.21 | 0.03 | 0.06 | −0.19 | 0.22 | 0 | 0.18 | 0 | 0.19 | 0.05 | 0.27 | 0.21 | 0.12 | −0.04 | 0.24 | 0.14 | 0.45 | 0.35 | 0.38 | 0.28 |
| crd2 | 0.29 | 0.16 | 0.21 | −0.06 | 0.29 | 0.16 | 0.32 | 0.17 | 0.37 | 0.21 | 0.4 | 0.38 | 0.25 | 0.15 | 0.28 | 0.26 | 0.37 | 0.33 | 0.35 | 0.29 |
| re1 | 0.47 | 0.27 | 0.31 | 0.24 | 0.22 | 0.13 | 0.31 | 0.27 | 0.37 | 0.22 | 0.31 | 0.29 | 0.24 | 0.04 | 0.07 | 0.05 | 0.12 | 0.09 | 0.05 | 0.1 |
| re2 | 0.18 | 0.1 | 0.43 | 0.35 | 0.17 | 0.13 | 0.14 | 0.2 | 0.09 | 0.33 | 0.21 | 0.11 | 0.2 | 0.17 | 0.08 | 0.01 | 0.05 | 0.11 | 0.05 | 0.07 |
| re3 | 0.21 | 0.08 | 0.18 | 0 | 0.22 | 0.29 | 0.27 | 0.3 | 0.05 | 0.12 | 0.37 | 0.16 | 0.28 | 0.18 | 0.26 | 0.22 | 0.08 | 0.14 | 0.27 | 0.2 |
| re4 | 0.09 | 0.13 | 0.15 | 0.07 | 0.17 | 0.22 | 0.31 | 0.29 | 0.17 | 0.14 | 0.13 | 0.32 | 0.21 | 0.15 | 0.23 | 0.24 | 0.02 | 0.03 | 0.01 | −0.05 |
| rfs1 | 0.03 | −0.01 | 0.03 | 0.05 | 0.14 | 0.08 | 0.14 | 0.08 | 0.12 | 0.26 | 0.26 | 0.14 | 0.3 | 0.25 | 0.37 | 0.26 | −0.24 | −0.13 | 0.09 | 0.04 |
| rfs2 | 0.16 | −0.03 | −0.05 | −0.02 | 0.03 | 0 | 0.13 | −0.04 | 0.02 | −0.04 | 0.03 | −0.02 | 0.01 | −0.06 | −0.05 | −0.08 | 0.1 | 0.08 | 0.14 | 0.17 |
| rfs3 | 0.05 | −0.02 | 0.08 | 0 | −0.02 | 0.04 | 0.06 | 0.07 | 0.1 | 0.17 | −0.06 | 0.13 | 0.16 | 0.23 | 0.13 | 0.17 | 0.11 | 0.01 | 0.04 | 0.07 |
| rfs4 | 0 | 0.08 | −0.13 | −0.09 | 0.04 | −0.01 | 0.2 | 0.12 | −0.07 | −0.11 | 0.01 | 0.02 | 0.02 | 0.09 | 0.03 | 0.04 | −0.04 | 0.06 | 0.11 | 0.25 |
| se1 | 1 | 0.65 | 0.42 | 0.31 | 0.4 | 0.27 | 0.44 | 0.15 | 0.59 | 0.3 | 0.31 | 0.11 | 0.28 | 0.1 | 0.11 | 0.06 | 0.39 | 0.27 | 0.28 | 0.24 |
| se2 | 0.65 | 1 | 0.2 | 0.29 | 0.12 | 0.19 | 0.2 | 0.18 | 0.55 | 0.33 | 0.08 | 0.08 | 0.15 | 0.04 | −0.1 | −0.07 | 0.14 | 0.05 | 0.03 | 0 |
| se3 | 0.42 | 0.2 | 1 | 0.53 | 0.27 | 0.24 | 0.07 | −0.05 | 0.33 | 0.59 | 0.31 | −0.02 | 0.14 | 0.31 | 0.08 | 0.05 | 0.21 | 0.21 | 0.13 | 0.23 |
| se4 | 0.31 | 0.29 | 0.53 | 1 | 0.05 | 0.22 | 0.08 | 0.16 | 0.14 | 0.48 | 0.09 | −0.02 | 0.04 | 0.25 | −0.07 | −0.08 | −0.06 | 0.03 | 0.02 | 0.04 |
| se5 | 0.4 | 0.12 | 0.27 | 0.05 | 1 | 0.59 | 0.48 | 0.15 | 0.23 | 0.25 | 0.68 | 0.24 | 0.25 | 0.12 | 0.32 | 0.12 | 0.29 | 0.18 | 0.29 | 0.24 |
| se6 | 0.27 | 0.19 | 0.24 | 0.22 | 0.59 | 1 | 0.29 | 0.26 | 0.1 | 0.26 | 0.49 | 0.09 | 0.15 | 0.09 | 0.18 | −0.02 | 0.11 | −0.01 | 0.11 | 0.11 |
| se7 | 0.44 | 0.2 | 0.07 | 0.08 | 0.48 | 0.29 | 1 | 0.65 | 0.26 | 0.17 | 0.42 | 0.48 | 0.18 | 0.01 | 0.23 | 0.16 | 0.29 | 0.22 | 0.39 | 0.33 |
| se8 | 0.15 | 0.18 | −0.05 | 0.16 | 0.15 | 0.26 | 0.65 | 1 | 0.13 | 0.15 | 0.2 | 0.42 | 0.05 | 0.01 | 0.18 | 0.1 | 0.1 | 0.05 | 0.12 | 0.05 |
| si1 | 0.59 | 0.55 | 0.33 | 0.14 | 0.23 | 0.1 | 0.26 | 0.13 | 1 | 0.51 | 0.37 | 0.36 | 0.5 | 0.25 | 0.27 | 0.27 | 0.24 | 0.17 | 0.19 | 0.16 |
| si2 | 0.3 | 0.33 | 0.59 | 0.48 | 0.25 | 0.26 | 0.17 | 0.15 | 0.51 | 1 | 0.43 | 0.23 | 0.33 | 0.39 | 0.16 | 0.15 | 0 | 0.05 | 0.15 | 0.13 |
| si3 | 0.31 | 0.08 | 0.31 | 0.09 | 0.68 | 0.49 | 0.42 | 0.2 | 0.37 | 0.43 | 1 | 0.37 | 0.37 | 0.23 | 0.44 | 0.34 | 0.14 | 0.08 | 0.29 | 0.27 |
| si4 | 0.11 | 0.08 | −0.02 | −0.02 | 0.24 | 0.09 | 0.48 | 0.42 | 0.36 | 0.23 | 0.37 | 1 | 0.32 | 0.12 | 0.24 | 0.47 | 0.2 | 0.09 | 0.38 | 0.33 |
| sn1 | 0.28 | 0.15 | 0.14 | 0.04 | 0.25 | 0.15 | 0.18 | 0.05 | 0.5 | 0.33 | 0.37 | 0.32 | 1 | 0.65 | 0.65 | 0.57 | 0.13 | 0.03 | 0.25 | 0.19 |
| sn2 | 0.1 | 0.04 | 0.31 | 0.25 | 0.12 | 0.09 | 0.01 | 0.01 | 0.25 | 0.39 | 0.23 | 0.12 | 0.65 | 1 | 0.48 | 0.53 | 0.01 | 0.07 | 0.14 | 0.05 |
| sn3 | 0.11 | −0.1 | 0.08 | −0.07 | 0.32 | 0.18 | 0.23 | 0.18 | 0.27 | 0.16 | 0.44 | 0.24 | 0.65 | 0.48 | 1 | 0.59 | 0.06 | 0.07 | 0.27 | 0.08 |
| sn4 | 0.06 | −0.07 | 0.05 | −0.08 | 0.12 | −0.02 | 0.16 | 0.1 | 0.27 | 0.15 | 0.34 | 0.47 | 0.57 | 0.53 | 0.59 | 1 | 0.13 | 0.04 | 0.27 | 0.16 |
| ta1 | 0.39 | 0.14 | 0.21 | −0.06 | 0.29 | 0.11 | 0.29 | 0.1 | 0.24 | 0 | 0.14 | 0.2 | 0.13 | 0.01 | 0.06 | 0.13 | 1 | 0.65 | 0.52 | 0.5 |
| ta2 | 0.27 | 0.05 | 0.21 | 0.03 | 0.18 | −0.01 | 0.22 | 0.05 | 0.17 | 0.05 | 0.08 | 0.09 | 0.03 | 0.07 | 0.07 | 0.04 | 0.65 | 1 | 0.56 | 0.45 |
| ta3 | 0.28 | 0.03 | 0.13 | 0.02 | 0.29 | 0.11 | 0.39 | 0.12 | 0.19 | 0.15 | 0.29 | 0.38 | 0.25 | 0.14 | 0.27 | 0.27 | 0.52 | 0.56 | 1 | 0.76 |
| ta4 | 0.24 | 0 | 0.23 | 0.04 | 0.24 | 0.11 | 0.33 | 0.05 | 0.16 | 0.13 | 0.27 | 0.33 | 0.19 | 0.05 | 0.08 | 0.16 | 0.5 | 0.45 | 0.76 | 1 |

## Appendix D. Code Used for Analysis

The code used for the SEM analysis can be found at the following GitHub page: https://github.com/TKForgeron/SEMinR-Analysis-The-Effect-of-Countermeasure-Readability-on-Security-Intentions (accessed on 14 September 2021).

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
