# Peer review of "The Effect of Countermeasure Readability on Security Intentions"

_jcp, doi:10.3390/jcp1040034_

Round 1

Reviewer 1 Report

Effect of Countermeasure Readability on Security Intentions

The paper argues that the readability of countermeasures influences security intentions and thus may enhance the security of a system. The authors rely on the Protection Motivation Theory (PMT) by Rogers in 1975. PMT proposes that people protect themselves based on the appraisal of two factors: threat and coping. It is applied to desktop security as shown in Fig 1. The approach relies on the behavioral views such as PMT, Self-Efficacy Theory of Motivation, Self-Determination Theory, and Health Belief Model. The paper discusses metrics for readability and then proposes a conceptual framework given in Fig 2. Fig 2 serves to identify 15 related hypotheses mostly based on existing literature as given in Table 1. The new hypotheses proposed are H5 Countermeasure readability is a positive predictor of self-efficacy, and H6 Countermeasure readability is a negative predictor of response cost. 

Based on these a questionnaire was developed and answers were sought from 131 volunteer respondents contacted personally or using a Whats App group. The results are presented in Figure 3. It shows that out of 15, 6 hypotheses were supported. Since H5 and H12 are supported, the paper concludes that  countermeasure readability affects security intentions through self-efficacy.

The main argument is fairly obvious, however a study supporting the view can be useful since it  stresses the need for better readability of security measures. 

A main limitation the paper is that it relies on perceptions of the respondents. It is not clear why their opinion is significant. What qualifications/experience do the have in the field? How do we know that they are not responding in the way they were supposed to respond?  Another limitation is that the work relies on what a technical-oriented reader may think of soft psychological theories.  

Author Response

Dear reviewer, 

Thank you for taking the time to assess our submission and for providing thoughtful comments. We agree that the inclusion of a study supporting the investigation of readability is useful. To address this point, we extended the sentence in lines 47-49 to reflect that readability positively influences people's understanding of information security policies. We supplemented the original reference at the end of the sentence with another reference investigating this topic.

We agree with the reviewer that our methods rely on the perceptions of reviewers, and that this forms a limitation of our paper. We address this limitation in our ‘Implications and Limitations’ section (6.1). We discuss the impact of using a high percentage of bachelor students in such a study (lines 553-558). Our methods and considerations align with earlier work by, for example, Hanus & Wu (2016) who use “a survey administered among undergraduate students” and Martens et al. (2019) who “made use of an online questionnaire.”

Regarding the question on participants simply responding in the way they were supposed to, this is a warranted concern. However, we did not inform participants whether they were viewing a countermeasure text that was less- or more difficult to read, meaning for this factor they could not have known how they were expected to respond. We describe how the process was implemented in the survey in lines 296-306.

Finally, we understand the reviewer's remark that we rely on behavioural theories in our work, which technically oriented readers may perceive as “soft psychological theories.” Nevertheless, we believe the study of human factors in security, as this special issue also looks to address, requires an appreciation by technically oriented researchers (including ourselves) for the body of work on behavioural theories. 

We hope that by extensively discussing the behavioural theories in Sections 2 and 3 and mentioning earlier work employing these theories in the security setting, we can convince readers of the relevance of these theories. We may be helped by the security community itself, as it is increasingly attentive towards human factors. This is witnessed by the popularity of symposia/conferences such as SOUPS (Symposium on Usable Privacy and Security) and HAISA (Human Aspects of Information Security & Assurance). 

We hope our answers and adaptations have addressed your comments adequately. Thank you once more for taking the time to assess our submission. 

Yours, 

The Authors

Reviewer 2 Report

Overall great work and a pleasure to read. Here are my suggestions on the actual manuscript as the research and insights are perfectly solid:

The phrase "Earlier work has shown the positive impact readability has on understanding and readability metrics make measuring and improving readability simple." on the abstract could probably be improved as it's hard to understand on a first pass.

References 2-5 don't refer SMEs specifically, and since it's not necessary for the body of work I recommend simply changing it to "companies and public institutions".

Reference 6 states $945 billion, to reach $1 trillion defense expense needs to be taken into account. Still as it's just context I suggest changing "over" to "just under".

Phrase on page 7 "Applied to our context meaning, cybersecurity threats are not being felt or seen." is understandable but it's another one that is hard to grasp in a first pass. Could be improved?

The "then" on line 307 is not necessary.

Phrase on page 9 "It is plain manifest variables are the basis of the measurement model, and therefore SEM analysis as a whole" should be redone for better readability. Might be as simple as changing "It is" to "Its".

Reference 42 seems to be incorrectly defined. The second list of names seem to be the editors but it's a bit much for a reference.

References [53, 83, 92-95, 97,98] could use an URL as some have very common phrasing that makes it hard to pinpoint that actual work

Author Response

Dear reviewer, 

Thank you for your kind words and for taking the time to read and assess our submission. We have made some adaptations to our submission based on your suggestions. We will walk through the adaptations one by one.

We agree with your remark regarding the sentence in our abstract which is difficult to understand. We modified the sentence to ‘Earlier work has shown the impact of readability on understanding and that readability metrics make measuring and improving readability simple’ (lines 9-10).

Thank you also for the input regarding references 2-5, we adapted it to ‘companies and public institutions’, to match your suggestion (line 24).

Similarly, you are right in correcting our statement regarding reference 6. We changed it to ‘just under’ on line 27.

Thank you for noticing the phrase on page 7, which certainly lacks some readability (no pun intended). We altered it to ‘Applied to our context this could mean that cybersecurity threats are not being felt or seen’ on lines 266-267.

Regarding your comment on the wording in line 307, we agree and have removed the ‘then’ on this line.

Thank you once again for offering suggestions to improve the readability of our text, we ended up choosing to remove the first words of the sentence, so that it now starts as ‘Manifest variables are...' (lines 391-392).

Reference 42 (now 43) indeed contained a list of editor names. We replaced the original reference with the standard BibTex reference as provided in Google Scholar, which does not contain the extensive editor list.

Thank you, finally, for the sharp observation regarding reference URLs. We went through all our references and added URLs where we deemed this necessary, of course including your specific suggestions.

To conclude, we would like to thank you once more for reading and assessing our submission. It is clear from your detailed remarks that you have thoroughly reviewed our paper. We sincerely appreciate your efforts.

We hope to have adequately addressed your comments.

Yours, 

The Authors

Reviewer 3 Report

Very well done. Only a minor comment: the related work sections should be highlighted more on why readability matrices are important for cyber security countermeasures and the lack of existing works in this domain.

Author Response

Dear reviewer,

Thank you for your kind words regarding our submission. We agree that additional focus can be placed on why readability metrics are important in our context and the lack of existing work in this area. We have made two changes to our submission to address this comment.

Firstly, we included an additional reference in our introduction and slightly extended our argument (lines 47-49), to highlight the relevance of the earlier work by Alkhurayyif and Weir (2017, 2018). Secondly, in our section on ‘Readability of Cybersecurity Countermeasures’ (2.2.3), we noted that earlier work in this domain is scarce (lines 192-193). We additionally extended our treatment of the earlier work by Alkhurayyif and Weir (2017, 2018), to provide the reader with more insight into why readability may be important for the understanding of cybersecurity countermeasures (lines 201-204).

We thank you once more for taking the time to review our submission. We hope to have adequately addressed your comments.

Yours, 

The Authors

Reviewer 4 Report

This article studies the effect of countermeasure readability on security Intentions. The paper is well organized.  I enjoyed reading it. My only recommendation for the authors is to add a section in their introduction to enumerate their paper’s contributions. and delete the sentences about their contributions within the body of their manuscript (Please refer to my comments on the document.)

Author Response

Dear reviewer,

Thank you for your kind comments and for reviewing our submission. We will first address the comments you mention explicitly in your feedback text, before turning to your comments in the PDF.

We understand the suggestion to add an explanation of our contribution to the introduction. We extended the paragraph presenting our main research question to contain an elaboration of our contributions (lines 59-62). We scanned the remaining document to delete unnecessary use of the word ‘contribute’ to avoid any confusion (line 355; line 443), but did not necessarily find redundant mentions of contributions in the body.

Regarding additional comments from the PDF, we have firstly corrected the sentence starting with “Applications often look to...” to “Application developers often look to” as per your suggestion (line 36). Thank you for this sharp insight.

Finally, we thank you for your comment on Figure 1, to avoid any potential copyright issues. However, although Hanus and Wu (2016) also include a Protection Motivation Theory (PMT) figure in their paper, we constructed our version to be clearly distinct. Hanus and Wu (2016), in turn, base their figure on earlier work in the PMT field. We nevertheless wanted to give appropriate credit to Hanus and Wu (2016), since they are the first to use this exact terminology.

We hope these alterations and explanations adequately address your comments. We thank you once more for the time you have invested in reviewing our submission.

Yours, 

The authors